# WHY SETTLE FOR ONE?
# TEXT-TO-IMAGESET GENERATION AND EVALUATION

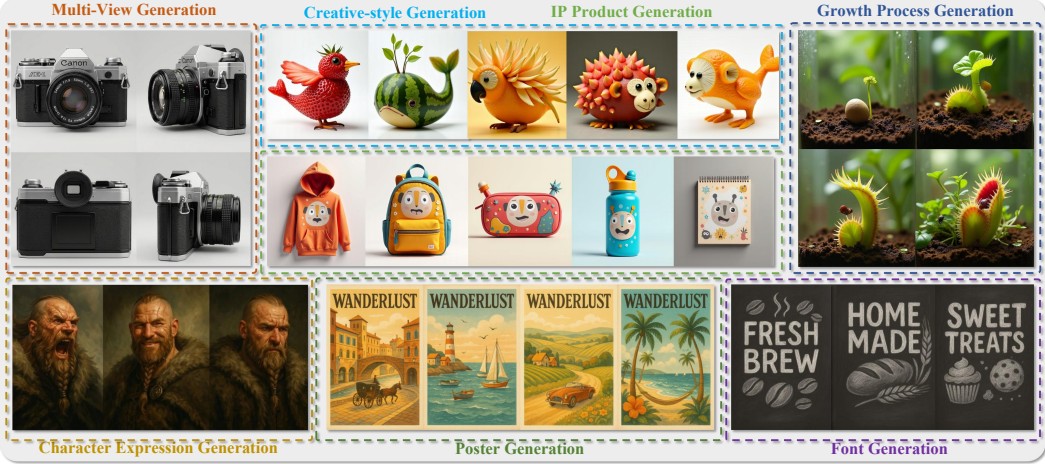

Figure 1: Illustration of diverse applications of Text-to-ImageSet (**T2IS**) generation. All examples are from the proposed **T2IS-Bench**. The proposed **AutoT2IS** method can generate diverse image sets (top row) and can also be seamlessly integrated with commercial models like GPT-4o (bottom row).

## ABSTRACT

Despite remarkable progress in Text-to-Image models, many real-world applications require generating coherent image sets with diverse consistency requirements. Existing consistent methods often focus on a specific domain with specific aspects of consistency, which significantly constrains their generalizability to broader applications. In this paper, we propose a more challenging problem, Text-to-ImageSet (T2IS) generation, which aims to generate sets of images that meet various consistency requirements based on user instructions. To systematically study this problem, we first introduce **T2IS-Bench** with 596 diverse instructions across 26 subcategories, providing comprehensive coverage for T2IS generation. Building on this, we propose **T2IS-Eval**, an evaluation framework that transforms user instructions into multifaceted assessment criteria and employs effective evaluators to adaptively assess consistency fulfillment between criteria and generated sets. Subsequently, we propose **AutoT2IS**, a training-free framework that maximally leverages pretrained Diffusion Transformers' in-context capabilities to harmonize visual elements to satisfy both image-level prompt alignment and set-level visual consistency. Extensive experiments on T2IS-Bench reveal that diverse consistency challenges all existing methods, while our AutoT2IS significantly outperforms current generalized and even specialized approaches. Our method also demonstrates the ability to enable numerous underexplored real-world applications, confirming its substantial practical value. All our data and code will be publicly available.

## 1 INTRODUCTION

Recent advancements in Text-to-Image (T2I) models (Esser et al., 2021; Rombach et al., 2022; Saharia et al., 2022; Betker et al., 2023; Baldridge et al., 2024; Labs, 2024; OpenAI, 2025) have substantially enhanced the ability to generate visually compelling and semantically faithful images. However, as illustrated in Figure 1, most practical scenarios, *e.g.*, product design, process illustrations,

or character creation, often require not a single image, but a coherent image set. Such image set must maintain ***set-level visual consistency*** with varying degrees of ***identity preservation*** (Huang et al., 2024a), ***uniform style*** (Hertz et al., 2024), and ***logical coherence*** (Song et al., 2025). This novel requirement poses huge challenges for current T2I models, which are primarily focused on image-level prompt alignment rather than set-level visual consistency.

Previous attempts addressing this coherence have largely concentrated on specific facets of consistency (Li et al., 2024a; Ye et al., 2023; Song et al., 2025) or specific combinations of consistency types (Mao et al., 2024; Chen et al., 2024). For instance, consistent character generation approaches (Tewel et al., 2024; Liu et al., 2025; Li et al., 2024a) aim to preserve character identity across various contexts, while methods such as StoryDiffusion (Zhou et al., 2024b) are tailored for generating stylistically coherent comic sequences with localized identity preservation. Despite promising progress within specific domains, they heavily rely on specialized data, techniques, and model architectures, suffering from additional resource and model design costs. More importantly, their focus on such specific aspects of visual coherence significantly limits their generalizability to a wider variety of applications. To this end, we propose a more challenging problem, ***Text-to-ImageSet (T2IS)*** generation, which aims to receive diverse user instructions and generate image sets that satisfy multifaceted consistency requirements. We address this challenge through two key contributions:

**(1) T2IS Benchmark:** We first present **T2IS-Bench**, the pioneering benchmark crafted specifically for T2IS generation. Derived from a comprehensive collection of real-world requirements, T2IS-Bench encompasses 596 representative tasks distributed across 26 detailed subcategories. Figure 1 showcases illustrative examples of diverse tasks. Based on this, we introduce **T2IS-Eval**, an evaluation framework that automatically transforms user instructions into multifaceted assessment criteria across three aspects of consistency: ***identity***, ***style***, and ***logic***. Our framework then leverages the strong multi-image recognition capabilities of large-scale models (Bai et al., 2025) to serve as effective consistency evaluators and obtain logits of "Yes-or-No" answers for each criterion to assess the consistency fulfillment. More importantly, unlike previous methods that focus on a single aspect of consistency (Hessel et al., 2021; Fu et al., 2023), T2IS-Eval leverages transformed criteria to enable adaptive and interpretable consistency evaluation.

**(2) T2IS Generation:** We propose **AutoT2IS**, a training-free framework that maximally leverages the remarkable in-context generation capabilities of pretrained Diffusion Transformers (DiTs) (Labs, 2024; Huang et al., 2024c). Specifically, AutoT2IS first employs a structured recaptioning approach to systematically parse user instructions into informative prompts for both individual images and the global image set. Subsequently, we introduce a novel set-aware generation with the divide-and-conquer strategy: first binding individual images' prompts to their respective independent latents during early denoising stages to establish unique content characteristics, then integrating these latents through the multi-modal attention mechanism with a global consistency prompt. This enables each visual latent to simultaneously attend to its individual prompt, the global consistency prompt, and other images' visual latents. This way dynamically harmonizes visual elements to satisfy both image-level prompt alignment and set-level visual consistency, effectively bridging the gap between isolated image generation and set-level consistency without requiring specialized fine-tuning.

We conduct comprehensive experiments on the proposed T2IS-Bench to evaluate various approaches, including unified models (*e.g.*, Show-o (Xie et al., 2024), Janus-Pro (Chen et al., 2025)), compositional frameworks (*e.g.*, Gemini+Flux (Labs, 2024)), and agentic frameworks (*e.g.*, ISG-Agent (Chen et al., 2024), ChatDiT (Huang et al., 2024d)). Our evaluations indicate that while current methods have achieved good performance on prompt alignment, there remains a significant gap in achieving set-level consistency, highlighting the value of this underexplored research direction. Furthermore, both quantitative and qualitative results demonstrate AutoT2IS substantially outperforms existing methods, achieving improvements across all aspects of consistency. AutoT2IS also achieves competitive or even superior performance in specific domains compared to specialized methods.

Interestingly, during the development of this work, advanced commercial image generation models like Gemini 2.0 (Google Cloud, 2025) and GPT-4o (OpenAI, 2025) were released. This positions T2IS-Bench as the first platform to systematically evaluate and analyze their T2IS capabilities. Our assessments reveal that these models excel in consistency but often sacrifice image quality and prompt alignment. To this end, as shown in the bottom of Figure 1, AutoT2IS can be seamlessly integrated with models like GPT-4o to further boost performance. By offering a transparent benchmark alongside a robust baseline, our work lays a strong foundation for the advancement of T2IS research.

## 2 RELATED WORK

**Text-to-Image Generation.** With the advancement of diffusion models (Ho et al., 2020; Song et al., 2020), Text-to-Image (T2I) models (Ramesh et al., 2021; 2022; Esser et al., 2021; Rombach et al., 2022; Saharia et al., 2022; Betker et al., 2023; Podell et al., 2023; Esser et al., 2024; Baldridge et al., 2024; Labs, 2024; OpenAI, 2025; Jia et al., 2025) have demonstrated exceptional capabilities in generating high-quality images that accurately align with textual descriptions. Early approaches, such as DALL-E 2 (Ramesh et al., 2022) and Stable Diffusion (Podell et al., 2023), employed Latent Diffusion Models (Rombach et al., 2022) (LDMs) with U-Net backbones (Ronneberger et al., 2015) for efficient denoising in compressed latent space. Recent developments have advanced the field through the integration of transformer architectures into diffusion models, known as Diffusion Transformers (DiTs) (Peebles & Xie, 2023; Esser et al., 2024; Labs, 2024). These models leverage global attention mechanisms to capture complex dependencies in data, resulting in improved scalability and image fidelity. However, existing methods primarily focus on generating individual images, with limited exploration into the generation of coherent image sets. In this paper, we investigate and harness the in-context generation capabilities of DiTs to maximize their potential for addressing T2IS tasks.

**Consistent Text-to-Image Generation.** Methods related to T2IS generation cover a wide range of domains, such as consistent character generation (Li et al., 2024a; Tewel et al., 2024; Liu et al., 2025) and visual storytelling (Mao et al., 2024; Zhou et al., 2024b). For character consistency, training-based approaches like PhotoMaker (Li et al., 2024a) employ parameter-efficient fine-tuning to preserve identity, while training-free methods such as One-Prompt-One-Story (Liu et al., 2025) leverage contextual knowledge in language models to enforce consistency without retraining. Storytelling, in addition to maintaining identity, often requires style coherence across images; for example, Make-a-Story (Rahman et al., 2023) uses a visual memory module for contextual alignment, and StoryDiffusion (Zhou et al., 2024b) employs Consistent Self-Attention to enhance consistency in both identity and style. However, for more complex tasks like process generation (Song et al., 2025), which demand logical consistency in visual progression, prior methods frequently require specialized task-specific fine-tuning and struggle to generalize (Song et al., 2025). This indicates that existing techniques are largely domain-specific.

**Interleaved Text-to-Image Generation.** Recent studies (Zhou et al., 2024a; Chen et al., 2024; An et al., 2023; Liang et al., 2024) have extended text-to-image research towards interleaved generation, in which multimodal sequences of alternating images and texts are produced to form coherent narratives or documents. These works have focused on linguistic-visual interplay, narrative coherence, and evaluation protocols designed for story flows. Although both interleaved generation and T2IS require multi-image reasoning, they differ in their settings and evaluation focus: interleaved generation prioritizes narrative consistency, text-image referential grounding, and temporal order, whereas T2IS generation directly synthesizes unordered image sets with an exclusive emphasis on visual consistency across identity, style, and logical progression. Furthermore, the proposed T2IS-Bench expands on existing benchmarks by covering a broader and more general set of consistency-centric subtasks, extending beyond the explicitly narrative-driven and interleaved scenarios studied in prior work.

## 3 METHODOLOGY

Our goal is to develop T2IS methods that aim to generate coherent image sets from diverse user instructions. To achieve this objective, we first provide a comprehensive T2IS benchmark with an evaluation framework for diverse tasks in Sec. 3.1. In Sec. 3.2, we introduce our generation method, AutoT2IS, which consists of Structured Recaption and Set-Aware Generation to generate image sets that satisfy both image-level prompt alignment and set-level visual consistency.

### 3.1 T2IS BENCHMARK

**T2IS-Bench.** We first construct a comprehensive benchmark that reflects real-world demands on diverse consistency, as illustrated in Figure 2. The construction follows a multi-stage process:

*First*, we conduct an extensive analysis of real-world T2IS applications and categorize these into five major groups, covering diverse consistency requirements. For instance, Character Generation primarily focuses on identity preservation, while Process Generation emphasizes logical consistency.

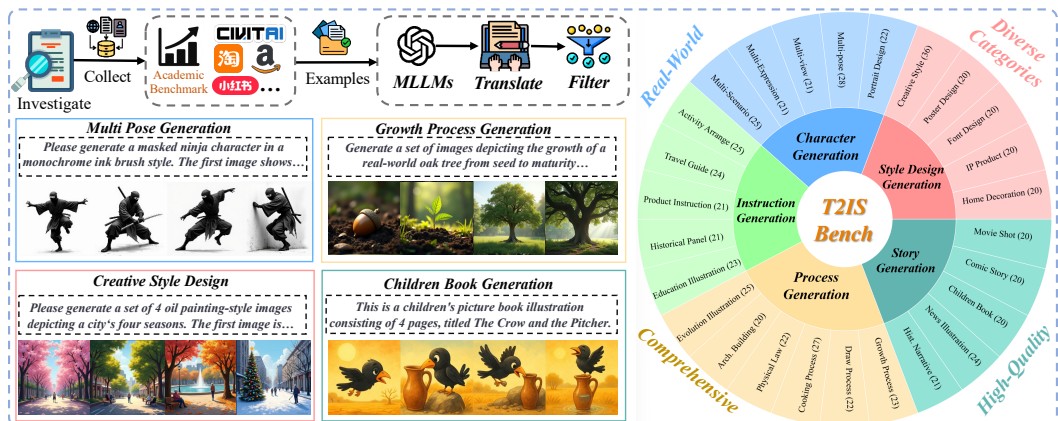

Figure 2: Illustration of the proposed T2IS-Bench. Upper Left: The collection process. Lower Left: Task examples displays. Right: Categories with their corresponding quantities.

It is noteworthy that falling within the same groups does not mean the requriements of consistency are exactly the same. Rather, categories within each group vary in the extent to which they integrate and balance additional dimensions of consistency.

*Second*, we collect diverse data from two complementary sources. We incorporate established benchmarks including Consistency+ (Liu et al., 2025), ISG-Bench (Chen et al., 2024), IDEA-Bench (Liang et al., 2024) and others (Song et al., 2025; Huang et al., 2024e;c). Additionally, we curate examples from real-world platforms, such as e-commerce websites (*e.g.*, Amazon, Taobao), image generation communities (*e.g.*, Civitai), social media platforms (*e.g.*, RedNote), and other sources. From academic benchmarks, we select representative examples based on their diversity, while from platforms, we prioritize examples with user engagement and popularity.

*Third*, we standardize all these examples by employing MLLMs such as GPT-4o to transform these into conversational user instructions. We further augment the dataset through in-context learning techniques (Dong et al., 2022) to ensure balanced samples across all subcategories. All samples undergo rigorous verification by PhD-level domain experts, with duplicate or highly similar instances eliminated through a combination of automated similarity detection using BertScore (Zhang* et al., 2020) and manual review.

Through the above process, we compile **T2IS-Bench**, containing 596 high-quality user instructions distributed across 26 distinct subcategories. Our benchmark offers the most comprehensive coverage to date, establishing a solid foundation for advancing T2IS. Detailed information regarding category definitions, representative examples, and dataset distribution statistics is provided in Appendix A.

**T2IS-Eval.** Although previous methods (Hu et al., 2023; Wiles et al., 2024; Hosseini et al., 2025) have adopted a **question-answering paradigm** to perform fine-grained dynamic evaluation, their criteria definition and execution are not suitable for T2IS tasks. Therefore, we propose **T2IS-Eval**, a comprehensive evaluation framework that assesses image sets across three critical consistency dimensions: *identity*, *style*, and *logic*. For each dimension, we employ LLMs to generate targeted evaluation questions as assessment criteria that reflect the consistency requirements corresponding to the user instructions. As illustrated in Figure 3, the criteria across different dimensions include:

- **Identity** criteria measure the consistency of identity preservation across images. This includes whether specific entities (*e.g.*, characters, objects) maintain consistent appearance features (*e.g.*, size, color, shape), and whether their emotional expressions or key identifying elements persist.

- **Style** criteria assess the style uniform across images, including consistency in illustration style (*e.g.*, watercolor, cartoon, 3D rendering), harmony in color palette usage, and others.

- **Logic** criteria evaluate whether the image set maintains reasonable causal relationships and narrative coherence. This includes environmental consistency across scenes, proper depiction of cause-and-effect relationships, and logical alignment between actions and their consequences.

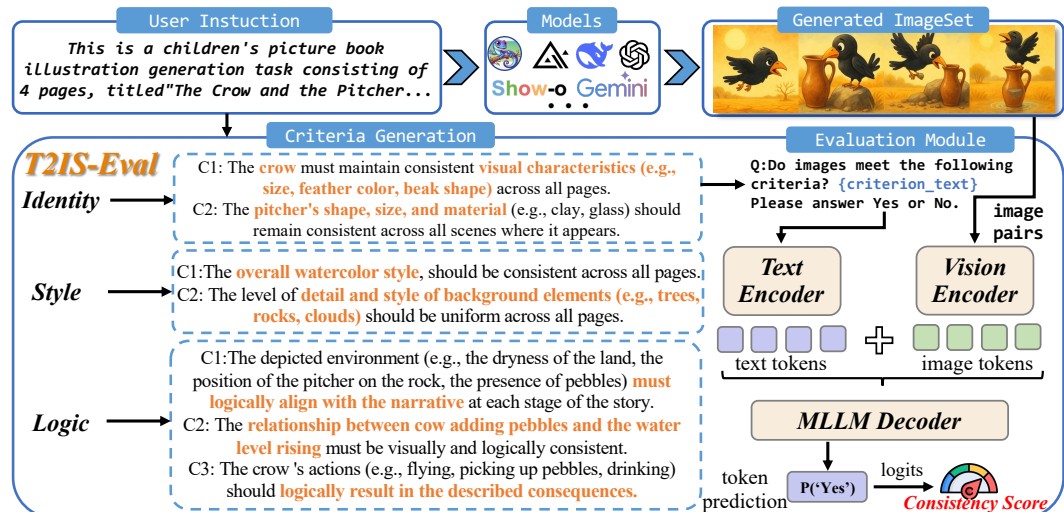

Figure 3: Overview of our T2IS-Eval. For each dimension, LLMs are prompted to generate 2-4 criteria; we only display a subset. Highlighted portions show the core elements for evaluation.

Then, we systematically assess how well image sets satisfy these criteria. Inspired by VQAscore (Lin et al., 2024), we discovered that large-scale MLLM (Bai et al., 2025) with strong multi-image recognition capabilities can serve as excellent consistency judges. Based on this insight, we formulate our evaluation criteria as questions that are sequentially applied to image pairs, obtaining logit scores for "Yes-or-No" answers that quantify the degree of consistency fulfillment. In Appendix B, we demonstrate the effectiveness of this approach and its advantages over direct scoring by MLLMs. Overall, unlike previous methods that focus on a single consistency dimension (Fu et al., 2023), our approach leverages transformed criteria to enables adaptive consistency evaluation with fine-grained measurements across multiple dimensions.

In addition to consistency evaluation, we also incorporate established methods to comprehensively assess generation quality. Specifically, we employ MPS (Zhang et al., 2024) to evaluate the aesthetic quality of generated images, and VQAscore (Lin et al., 2024) to measure the alignment between each image and its corresponding instruction. Notably, all three evaluation components (aesthetics, prompt alignment, and visual consistency) leverage vision-language models and employ a unified scoring mechanism based on next-token prediction logits or text-image matching logits. Consequently, these quantitative metrics are normalized to a consistent range $[0, 1]$, providing better interpretability and holistic assessment.

## 3.2 T2IS GENERATION

In this section, we introduce AutoT2IS, a training-free framework for T2IS generation. We aim to maximally leverage the remarkable in-context generation capabilities (Huang et al., 2024c) of pretrained Diffusion Transformers (DiTs (Peebles & Xie, 2023; Labs, 2024)) to handle diverse T2IS tasks. We delineate two key steps, as depicted in Figure 4. Specifically, AutoT2IS first utilizes a *structured recaption* to parse user instructions into informative prompts for both individual images and the complete image set, thereby capturing richer semantic information. Subsequently, we introduce *set-aware generation* that effectively applies the recaptioned individual prompts and consistency requirements to the entire image set, transforming semantic contents into visually consistent outputs.

**Structured Recaption.** Let $y$ denote the given user instruction, which contains requirements for individual images and overall consistency. We first employ LLMs to identify key content for each image and consistency requirements for the entire set. The recognized elements are represented in a structured format $S = \{E, C\}$, where $E = \{e_1, e_2, ..., e_n\}$ represents the content for $n$ individual images, and $C = \{c_1, c_2, ..., c_m\}$ denotes the $m$ consistency requirments extracted from $y$.

Based on the structured information $S$ and original user instruction $y$, we generate enhanced prompts for each image and global consistency requirements:

$$p_i = f_{\text{recap}}(e_i, y) \ \forall i \in \{1, 2, ..., n\}, \quad g = f_{\text{consist}}(C, y) \tag{1}$$

where $p_i$ represents the detailed prompt for the $i$-th image with denser fine-grained details to improve fidelity, and $g$ represents the global consistency description that ensures the model adheres to overarching principles across all images. Functions $f_{\text{recap}}$ and $f_{\text{consist}}$ are implemented using LLMs to expand the structured information into comprehensive textual descriptions.

**Set-Aware Generation.** While DiT models exhibit powerful in-context capabilities, existing approaches (Huang et al., 2024c;b; Tan et al., 2024) often require fine-tuning to activate these abilities, which compromises the model's original generalization performance. As shown in Figure 4, we introduce set-aware generation that enables consistency requirements to be expressed through textual prompts and multi-modal attention in a training-free way.

We empirically find that these few binding steps are sufficient to establish the unique content of each image, allowing subsequent steps to focus on resolving consistency issues across the image set. Inspired by this, we design a *divide-and-conquer* strategy to separate DiT's native denoising capabilities into two distinct phases. In the *divide* step, which is applied during the early stages of the denoising process with a total timestep $t$, we generate independent noise latents $x_t^i$ for each image $i \in \{1, 2, ..., n\}$ and denoise them separately with their corresponding prompts $p_i$. This binding process is executed only during the initial $r$ steps of the denoising process.

In the *conquer* phase, we concatenate all independently denoised latents $x_{t-r}^i$ into a grid layout $X_{t-r} = [x_{t-r}^1, x_{t-r}^2, ..., x_{t-r}^n]$. This concatenation enables previously isolated regions to collaborate through DiT's multi-modal attention mechanism (Peebles & Xie, 2023; Esser et al., 2024). Formally, for each denoising step $t' \in \{t - r, t - r - 1, ..., 0\}$, the attention computation can be expressed as:

$$\text{Attention}(Q, K, V, M) = \text{softmax}\left(\frac{QK^T + M}{\sqrt{d_k}}\right)V \tag{2}$$

where $Q = W_Q X_{t'}$ represents queries derived from the concatenated latent $X_{t'}$, $K = [W_K^{\text{text}}p; W_K^{\text{text}}g; W_K^{\text{image}}X_{t'}]$ and $V = [W_V^{\text{text}}p; W_V^{\text{text}}g; W_V^{\text{image}}X_{t'}]$ index keys and values from both the prompt embeddings $P = [p_1, p_2, ..., p_n, g]$ and the latent representation itself. The mask matrix $M \in \mathbb{R}^{N \times (N_p + N_g + N)}$ is defined as:

$$M_{i,j} = \begin{cases} 0, & \text{if } i \in x_{t'}^k \text{ and } j \in \{p_k, g, X_{t'}\} \\ -\infty, & \text{otherwise} \end{cases}, \quad \text{where } k \in \{1, 2, ..., n\}. \tag{3}$$

This mask ensures that each visual latent from a specific region $x_{t'}^k$ to simultaneously attend to three critical elements, as illustrated in Figure 4:

- Its corresponding individual prompt $p_i$ for precise semantic content preservation;
- The global consistency prompt $g$ for harmonious coherence across the image set;
- Visual latents in other images for enhanced cross-image alignment and contextual integration.

Through this enhanced multi-modal attention mechanism, the model dynamically harmonizes visual elements to satisfy both local alignment and global consistency requirements, effectively bridging the gap between isolated image generation and set-level consistency without specialized fine-tuning.

## 4 EXPERIMENTS

**Implementation Details.** We utilize DeepSeek-R1 (DeepSeek-AI, 2025) as the foundational LLM for criteria generation and structured recaptioning. Qwen2.5-VL-7B (Bai et al., 2025) serves as our consistency evaluator. For Set-Aware generation, we employ FLUX.1-dev (Labs, 2024) as the base model, executing a total of 20 denoising steps, i.e., 2 steps for the divide phase and the remaining for the conquer phase. For the concatenated latent, we apply positional encoding at new positions to preserve the original multi-modal attention structure.

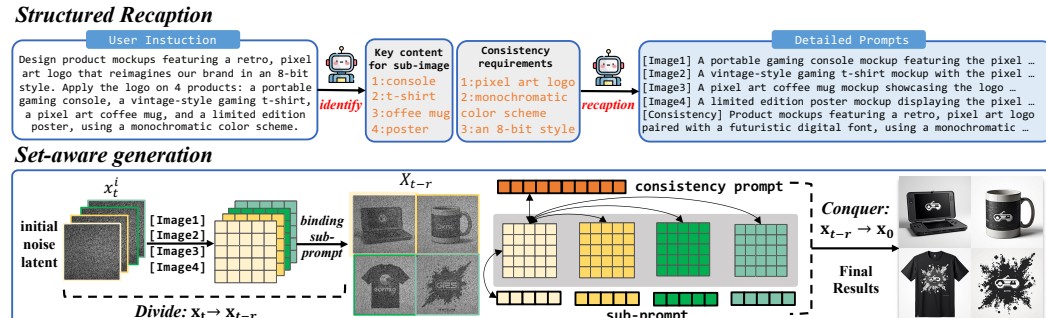

Figure 4: The pipeline of our AutoT2IS, consists of structured recaption and set-aware generation.

Table 1: Comparisons with various generalized generation models on T2IS-Bench. The upper shows open-source models, while the lower presents commercial models. The best and second-best results among the open-source models are highlighted in bold and underlined, respectively.

| Model | Aesthetics | A Prompt Alignment | | | 🖼 Visual Consistency | | | Avg. |
|---|---|---|---|---|---|---|---|---|
| | | Entity | Attribute | Relation | Identity | Style | Logic | |
| 👤 Anole (Chern et al., 2024) | 0.170 | 0.534 | 0.611 | 0.570 | 0.115 | 0.148 | 0.090 | 0.264 |
| 👤 Show-o (Xie et al., 2024) | 0.206 | 0.780 | 0.785 | 0.776 | 0.233 | 0.287 | 0.285 | 0.409 |
| 👤 Janus-Pro (Chen et al., 2025) | 0.333 | 0.787 | 0.785 | 0.781 | 0.224 | 0.272 | 0.300 | 0.435 |
| 👥 Gemini & SD3 (Esser et al., 2024) | 0.500 | **0.796** | **0.794** | **0.789** | 0.287 | 0.244 | 0.320 | 0.480 |
| 👥 Gemini & Pixart (Chen et al., 2023) | 0.447 | 0.743 | 0.765 | 0.747 | 0.206 | 0.279 | 0.268 | 0.440 |
| 👥 Gemini & Hunyuan (Li et al., 2024b) | 0.410 | 0.758 | 0.774 | 0.765 | 0.197 | 0.276 | 0.271 | 0.436 |
| 👥 Gemini & Flux-1 (Labs, 2024) | **0.533** | 0.791 | 0.790 | 0.786 | 0.249 | 0.302 | 0.328 | 0.490 |
| 🤖 ChatDit (Huang et al., 2024d) | 0.414 | 0.717 | 0.726 | 0.726 | 0.296 | 0.326 | 0.310 | 0.455 |
| 🤖 ISG-Agent (Chen et al., 2024) | 0.256 | 0.667 | 0.703 | 0.682 | 0.146 | 0.178 | 0.163 | 0.338 |
| 🤖 AutoT2IS (ours) | 0.520 | 0.729 | 0.756 | 0.743 | **0.359** | **0.414** | **0.356** | **0.515** |
| 👤 **Gemini 2.0 Flash** (Google Cloud, 2025) | 0.430 | 0.738 | 0.747 | 0.743 | 0.428 | 0.383 | 0.392 | 0.509 |
| 👤 **GPT-4o** (OpenAI, 2025) | 0.445 | 0.663 | 0.683 | 0.693 | 0.400 | 0.463 | 0.383 | 0.501 |
| 🤖 **AutoT2IS + GPT-4o** | 0.567 | 0.754 | 0.761 | 0.763 | 0.441 | 0.520 | 0.416 | 0.571 |

Due to space limitations, we present the most significant experimental results in the following, including quantitative comparison of both generalized and specialized methods with ours on T2IS-Bench (Sec. 4.1), qualitative visual comparisons to demonstrate the broad applications of our method (Sec. 4.2), and ablation studies on core components (Sec. 4.3). Please refer to the appendix for more comprehensive information. We provide detailed experimental settings in App.C, reliability analysis of T2IS-Eval in App.D, failure cases in App.G, limitations and future work in App.F.

## 4.1 BENCHMARKING T2IS: QUANTITATIVE COMPARISON OF GENERALIZED AND SPECIALIZED METHODS

**Experiment Setups.** We evaluate diverse frameworks for T2IS generation capabilities: (1) 👤 unified models like Show-o; (2) 👥 compositional models that use Gemini (Team et al., 2023) to generate sub-captions with state-of-the-art T2I models; and (3) 🤖 agentic models such as ChatDit that incorporate multi-step planning and generation processes. We also evaluate commercial models including Gemini 2.0 Flash and GPT-4o. Detailed setups for implementing these models are provided in Appendix K.

**Visual Consistency Challenges All Generalized Methods.** As illustrated in Table 1, most models perform well on prompt alignment, with average scores all exceeding 0.5, demonstrating the significant progress T2I models have made in alignment capabilities. However, all models exhibit significant deficiencies in visual consistency. All open-source methods score below 0.35 across all consistency dimensions, highlighting the value of exploring this research direction. Comparatively, our method achieves the best consistency capabilities among open frameworks, showing notable improvements over previous methods and delivering the best overall performance.

**Commercial Models Still Need Improvement.** As shown in the bottom of Table 1, we evaluate state-of-the-art commercial models (Google Cloud, 2025; OpenAI, 2025). These models, benefiting from large-scale pretraining, demonstrate superior instruction understanding and better consistency

Table 2: Comparisons of specialized methods and our unified AutoT2IS across different domains.

| Model | Aesthetics | A Prompt Alignment | | | Visual Consistency | | | Avg. |
|---|---|---|---|---|---|---|---|---|
| | | Entity | Attribute | Relation | Identity | Style | Logic | |
| *Character Generation: Multi-view, Multi-scenario, Portrait Design* | | | | | | | | |
| PhotoMaker (Li et al., 2024a) | 0.413 | 0.793 | 0.799 | 0.762 | 0.482 | 0.458 | 0.450 | 0.552 |
| OnePrompt (Liu et al., 2025) | 0.600 | 0.805 | 0.807 | 0.766 | 0.520 | 0.512 | 0.470 | 0.590 |
| X-Flux (AI, 2024) | 0.428 | 0.834 | 0.804 | 0.780 | 0.531 | 0.529 | 0.521 | 0.553 |
| **AutoT2IS (ours)** | 0.557 | 0.798 | 0.788 | 0.764 | 0.609 | 0.520 | 0.509 | **0.619** |
| *Style Design Generation: Creative Style, Font, IP product design* | | | | | | | | |
| IPAdapter-Flux (Team, 2024) | 0.514 | 0.856 | 0.833 | 0.846 | 0.386 | 0.455 | 0.517 | **0.582** |
| **AutoT2IS (ours)** | 0.475 | 0.789 | 0.791 | 0.792 | 0.339 | 0.391 | 0.478 | 0.533 |
| *Story Generation: Movie Shot, Comic Story, Children Book* | | | | | | | | |
| ICLora-Story (Huang et al., 2024c) | 0.408 | 0.711 | 0.739 | 0.702 | 0.078 | 0.190 | 0.119 | 0.361 |
| StoryDiffusion (Zhou et al., 2024b) | 0.581 | 0.684 | 0.728 | 0.670 | 0.182 | 0.207 | 0.141 | 0.413 |
| Story-Adapter (Mao et al., 2024) | 0.578 | 0.749 | 0.766 | 0.734 | 0.181 | 0.328 | 0.226 | **0.463** |
| **AutoT2IS (ours)** | 0.534 | 0.629 | 0.698 | 0.660 | 0.262 | 0.405 | 0.237 | 0.456 |
| *Process Generation: Growth, Draw, Building* | | | | | | | | |
| MakeAnything (Song et al., 2025) | 0.358 | 0.636 | 0.665 | 0.652 | 0.329 | 0.340 | 0.222 | 0.408 |
| **AutoT2IS (ours)** | 0.578 | 0.727 | 0.754 | 0.739 | 0.314 | 0.346 | 0.274 | **0.493** |

than open-source alternatives. However, our results reveal that they still struggle with T2IS generation tasks. During T2IS generation, their aesthetic quality and prompt alignment significantly deteriorate, even performing worse than many open-source models. This shortcoming can be mitigated by incorporating Auto-T2IS's structured recaption to enrich semantic information, which achieves significant improvements in quality, alignment, and consistency. These findings underscore the necessity for improved T2IS task instruction comprehension in commercial models.

**AutoT2IS Matches/Exceeds Specialized Methods.** Table 2 benchmarks our unified generation approach against domain-specific specialized methods. While specialized baselines are extensively fine-tuned for particular tasks, yielding strong performance in domains such as Style Design and Story Generation, our results demonstrate that AutoT2IS remains highly competitive across all categories, despite operating in a training-free manner. By avoiding domain-specific overfitting, AutoT2IS preserves broad generative robustness and adaptability, underscoring the strengths of a unified framework for diverse T2IS tasks.

## 4.2 VISUALIZING T2IS: QUALITATIVE COMPARISON AND DIVERSE APPLICATION SHOWCASE

**AutoT2IS Excels at Common T2IS Tasks.** Figure 5 presents a qualitative comparison between our AutoT2IS and existing methods across several common T2IS tasks. As demonstrated, generalized methods like Show-o and Flux exhibit significant limitations in identity preservation, style uniform, and logical coherence. In contrast, AutoT2IS produces more visually consistent image sets. When evaluated against specialized methods in multi-scenario character generation, our approach maintains exceptional identity consistency while ensuring precise prompt alignment, achieving results comparable to the SOTA method 1Prompt1Story (Liu et al., 2025). Notably, when it comes to challenging logical consistency generation tasks, existing methods like MakeAnything (Song et al., 2025), despite being specifically trained for these scenarios, often compromise their fundamental generation capabilities, resulting in diminished image quality. Conversely, our training-free method preserves the model's original generation capabilities and simultaneously maximizing its inherent consistency potential.

**AutoT2IS Enables More Real-world Applications.** Beyond common T2IS tasks, our AutoT2IS enables to address previously underexplored applications with significant real-world value, as illustrated in Figure 6. These applications have been largely overlooked due to the lack of a unified approach to varying degrees of different consistency. For instance, IP Product Design (Task 2) requires coherence between IP concept identity and product style. Similarly, Growth Process Generation and Physical Law Illustration (Tasks 4 and 6) demand comprehensive consistency across identity, style, and logical dimensions. The diversity of the accomplished tasks demonstrates that our Auto-T2IS effectively

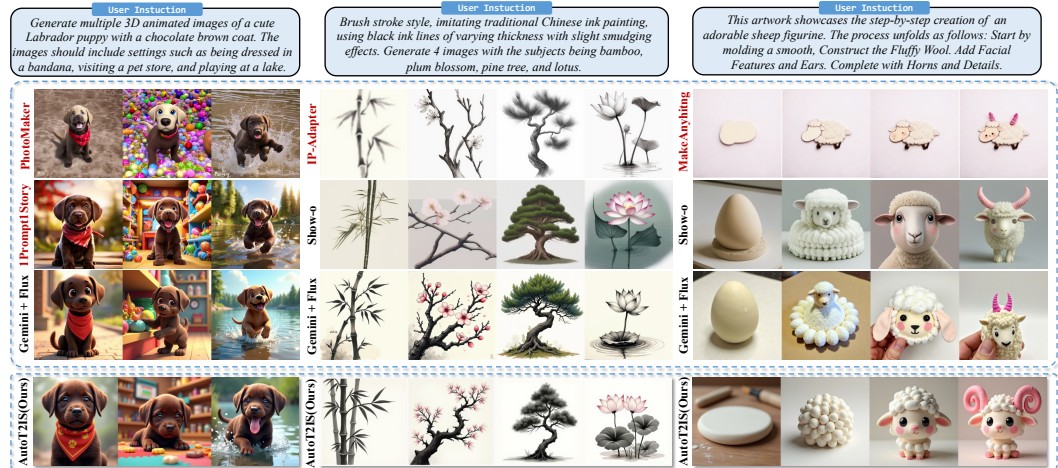

Figure 5: Qualitative comparison between our AutoT2IS and existing methods on common T2IS tasks. Black labels indicate generalized methods, while red labels represent specialized methods.

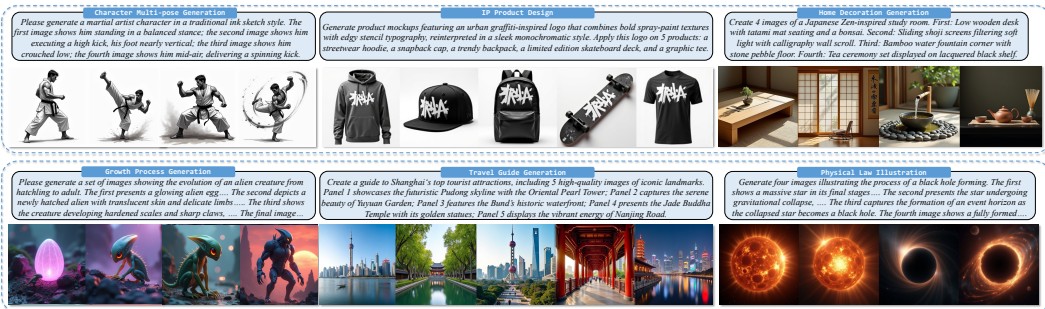

Figure 6: AutoT2IS enables novel T2IS applications with significant real-world value.

addresses diverse consistency challenges within a unified model, highlighting its potential for broad applicability across various T2IS practical scenarios.

## 4.3 ABLATION STUDIES

To better understand the contribution of each stage in AutoT2IS, we conduct ablation studies. Table 3 presents the quantitative results across different evaluation dimensions.

**Importance of Structured Recaption (SR).**
The *structured recaption* plays a critical role by reformulating user inputs into more structured and detailed descriptions that better guide the image generation process. When we replace the *structured recaption* with a baseline approach that simply uses an LLM to generate sub-captions, we observe a significant drop across all evaluation metrics. These results highlight the importance of semantic completeness and relational coherence for effective T2IS generation.

Table 3: Ablation studies on key stages.

| Variants | Aesthetics | Alignment | Consistency |
|---|---|---|---|
| Full | **0.520** | **0.742** | **0.376** |
| w/o SR | 0.512 | 0.695 | 0.241 |
| w/o SG | 0.326 | 0.611 | 0.318 |

**Importance of Set-Aware Generation (SG).** The set-aware generation is responsible for creating a consistent set across multiple images, while ensuring the quality and alignment of each sub-image. We replace this stage with a baseline approach that directly concatenates prompts to guide the model in generating multi-grid images (Huang et al., 2024c;d). As shown in results, despite achieving reasonable consistency, the quality of individual images and text-image alignment significantly decreased. This demonstrates the importance of the divide-and-conquer approach in Set-Aware Generation, where the model first understands the content of individual image before rendering them

with visual consistency. We include additional ablation with visualizations in Appendix J for more intuitive understanding.

## 5 CONCLUSION

In this paper, we advance the field of Text-to-ImageSet (T2IS) by introducing innovative frameworks for both T2IS generation and evaluation. We present T2IS-Bench, a comprehensive benchmark featuring 596 representative tasks across 26 detailed subcategories. Alongside this, we introduce T2IS-Eval, an evaluation framework that enables diverse consistency assessment. Furthermore, we introduce AutoT2IS, a training-free framework that fully exploits the exceptional in-context generation capabilities of DiT models, effectively harmonizing visual elements to meet both image-level prompt alignment and set-level visual consistency. By providing a transparent benchmark and a robust baseline, our work establishes a solid foundation for the future of T2IS research.

### FUTURE WORK

An exciting direction for future work is to leverage the capabilities of T2IS-Bench, AutoT2IS, and T2IS-Eval for directly training generalized T2IS models. As discussed, T2IS-Bench enables large-scale synthesis of high-quality (instruction, image-set) pairs for supervised finetuning (SFT), while T2IS-Eval offers fine-grained, multi-dimensional reward signals for reinforcement learning (RL) based optimization. Integrating these resources into a training workflow, where synthetic data drives SFT and automated verification enables RL reward modeling, has strong potential to advance the generalization and consistency capabilities of T2IS models. We plan to explore this "training via synthesis and verification" loop in future research, including systematic methods of RL and SFT paradigms, and building scalable models capable of robust consistency across a wide range of real world scenarios.

### ETHICS STATEMENT

This work focuses on text-to-imageSet generation. All datasets used in our experiments were carefully examined to prevent discrimination, privacy violations, or fairness concerns, and contain no personally identifiable information. Our contribution is intended to augment human creativity by alleviating repetitive design tasks rather than replacing professional creators. The proposed models are developed for responsible research purposes, and we encourage future studies to further investigate fairness and broader social impacts in generative systems.

### REPRODUCIBILITY STATEMENT

To ensure reproducibility, we provide all experimental details in Section 4 and Appendix C and K. All source code will be made public.

### LLM USAGE STATEMENT

We primarily used LLMs to assist with language polishing of the paper.

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

This appendix provides comprehensive supplementary materials to support our main paper.

> **Note:** We also provide an anonymous static website showcasing extensive generation demos. We encourage readers to explore our results at `./index.html`. The website will be further enhanced and made public with all code and data.

Below is a brief overview of each section:

- **Details of T2IS-Bench** (Appendix. A)
  Presents the construction and analysis of our benchmark dataset, including group definitions, dataset statistics.

- **Details of T2IS-Eval** (Appendix. B)
  Comprehensive details of our evaluation framework for assessing visual consistency, prompt alignment, and aesthetic quality in generated image sets, along with thorough comparisons against existing benchmarks and evaluations.

- **Details of Experimental Settings** (Appendix. C)
  Provides comprehensive technical details about model configurations, grid layouts, and parameters settings in our experiments.

- **Reliability Analysis of T2IS-Eval** (Appendix. D)
  Presents empirical analysis validating the reliability and effectiveness of our evaluation framework.

- **Ablation on Different Image Generation Backbones** (Appendix. E)
  Provides comprehensive details about the ablation study on different image generation backbones, including open-source and closed-source models.

- **Limitations and Future Work** (Appendix. F)
  Discusses current limitations of our approach and outlines promising directions for future research.

- **Analysis of Failure Cases** (Appendix. G)
  Analyzes typical failure cases and challenges for T2IS generation.

- **Analysis of Challenge Categories** (Appendix. H)
  Provides an in-depth analysis of the various challenge categories encountered in T2IS generation.

- **More Results for Reliability Analysis** (Appendix. I)
  This section provides additional results for the reliability analysis of our evaluation framework. It includes a statistical significance evaluation to determine the robustness of our metrics and an analysis of LLM performance and reliability in the context of T2IS generation.

- **Visualizations for Ablation Studies** (Appendix. J)
  Provides detailed visualizations and analysis of ablation studies to analyze the effectiveness of key components in our framework.

- **Implementation Details of Existing Methods** (Appendix. K)
  Describes implementation details of baseline methods used for comparison, including compositional, unified, agentic and commercial models.

- **Visualization Gallery** (Appendix. L)
  Provides a comprehensive visualization gallery showcasing generation results across different categories. For each category, we present detailed task descriptions, example instructions, and AutoT2IS-generated image sets to demonstrate our method's capabilities in handling diverse T2IS generation scenarios.

## A  MORE DETAILS OF T2IS-BENCH

We first present the categorization of each group. Then, we provide comprehensive statistical information about the entire dataset.

### A.1  GROUP DEFINITIONS

We construct our benchmark by carefully curating tasks from five groups that comprehensively evaluate different aspects of visual consistency:

**Character Generation.** This group focuses on maintaining consistent character identity across diverse contexts. Models must generate characters with consistent physical attributes (facial features, body proportions, clothing), and distinguishing characteristics even when shown in different poses, viewing angles, expressions, and environments.

**Design Style Generation.** This group evaluates the ability to maintain a cohesive visual style and aesthetic across different design elements and contexts. Models must focus on creating a unified visual language that ties different design pieces together, *e.g.*, posters, product packaging, brand materials, or decor elements, while ensuring the style remains distinctive and purposeful.

**Story Generation.** This group requires generating image sets that convey clear story progression while maintaining consistent characters and visual style. Tasks span movie scenes, comic panels, children's books, and news illustrations.

**Process Generation.** This group emphasizes temporal and causal consistency in depicting step-by-step procedures. Models must generate logically ordered sequences illustrating processes like growth, construction, cooking, or physical phenomena. Each sequence should maintain consistent visual style and recognizable elements across steps.

**Instruction Generation.** This group evaluates logical consistency in educational and instructional content. Tasks involve creating image sets that effectively communicate procedures, historical information, product usage, or travel guidance. Models must ensure consistent visual presentation, clear identification of elements, and logical sequencing.

In the Appendix L, we provide detailed descriptions and examples for each category in each group.

## A.2 DATASET STATISTICS

Our T2IS-Bench dataset contains a diverse collection with varying image set sizes and instruction lengths. As shown in Figure 7, the distribution of tasks exhibits several notable characteristics:

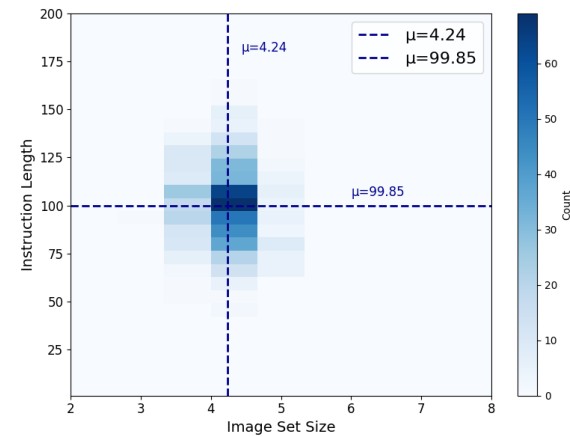

Figure 7: Distribution of T2IS-Bench in terms of image set size and instruction length. The heatmap shows the density of tasks, with darker blue indicating higher concentration.

**Image Set Size Distribution:** While tasks range from 2-8 images per set, we focus primarily on sets requiring 3-5 images, with a mean of 4.24 images. This range provides sufficient complexity for evaluating consistency while remaining computationally tractable. Tasks with 2 images are too simple to meaningfully assess consistency, while those above 5 images become overly complex, so we concentrate the distribution around 3-5 images per set with a peak at 4 images.

**Instruction Length Distribution:** The instructions range from 20 to 175 words, with a mean length of 99.85 words. Shorter instructions (20-50 words) provide flexibility in image generation by specifying only key requirements, while longer instructions (150-175 words) offer more precise control over visual elements and relationships. To balance flexibility and control, most tasks have instructions between 50-150 words, allowing sufficient detail to guide generation while avoiding over-specification.

This distribution ensures our benchmark provides a comprehensive evaluation of models' capabilities in handling both simple and complex T2IS generation tasks.

## A.3 DATASET CURATION PROCESS

Our dataset underwent a meticulous curation process to ensure high-quality and diverse instructions, comprising three main phases:

**1. Collection Guideline:** We first designed a standardized collection guideline to ensure instruction quality and diversity. It includes:

- Clear definitions of task types and target outcomes.

- Instruction templates and examples covering various identity, style, and logic dimensions.

- A checklist to ensure that each instruction requires non-trivial generation beyond simple captioning.

Each instruction and its category label were double-reviewed by PhD-level experts, with conflicts adjudicated by an additional expert.

**2. Sampling Procedure:** To ensure broad coverage across 26 generation categories, we applied stratified sampling:

- Starting from a large pool of open-ended instructions, we selected over 650 candidates.

- We ensured varying complexity (simple, moderate, abstract) across and within categories.

- Categories demanding high-level logical reasoning (e.g., story or process) were intentionally oversampled for robustness.

**3. Human Feedback and Refinement:** Finally, a Refinement Phase was conducted where experts reviewed model outputs for each instruction, evaluating:

- Clarity and executability.

- Factual correctness.

- Avoidance of overly long or NSFW content.

Low-quality instructions were revised or discarded. Feedback from early T2IS-Bench users further guided refinements. The final dataset includes 596 rigorously curated instructions.

## B    MORE DETAILS OF T2IS-EVAL AND COMPARISON WITH EXISTING BENCHMARKS

### B.1    MORE DETAILS OF T2IS-EVAL

**Visual Consistency Evaluation.** We utilize DeepSeek-R1 (DeepSeek-AI, 2025) as the foundational LLM for criteria generation, and Qwen2.5-VL-7B (Bai et al., 2025) serves as our consistency evaluator. Specifically, we first resize all images in the image set to $512\times512$ pixels. Then, we pair the images sequentially and feed each pair to Qwen2.5-VL-7B with generated criteria. For each pair, we compute the logits for "Yes" and "No" responses, apply softmax to obtain probabilities, and use the probability of "Yes" as the consistency score. The final score for each dimension is calculated by averaging the scores across all pairs within that dimension.

Compared to simple answer tokens, answer likelihoods provide more fine-grained measurements while avoiding potential uncertainty and hallucination issues that may arise from directly asking for numerical scores (Lin et al., 2024). In Appendix D, we further validate through experiments the effectiveness of using Qwen2.5-VL-7B as a consistency evaluator and demonstrate the advantages of this approach.

**Prompt Alignment Evaluation.** For evaluating prompt alignment capabilities, we adopt VQAScore (Lin et al., 2024), a well-established method for assessing image-text alignment. Specifically, we employ DeepSeek-R1 (DeepSeek-AI, 2025) to generate alignment criteria for each sub-image from three perspectives: Entity, Attribute, and Relation. We then evaluate each sub-image against these criteria and compute the average score. This approach effectively assesses whether T2IS models can maintain strong image-level prompt alignment capabilities when generating multiple coherent images.

**Aesthetic Quality Evaluation.** We further employ MPS (Zhang et al., 2024) to evaluate the overall aesthetic quality of generated images. MPS introduces a preference condition module built upon the CLIP model to learn diverse aesthetic preferences, including composition, light contrast, color

Table 4: Comparison with existing consistent generation benchmarks.

| | Instruction (# Numbers) | Category (# Numbers) | Category Coverage | Sub-image Evaluation? | Consistency Evaluation? | Adaptive Evaluation? |
|---|---|---|---|---|---|---|
| Consistory+ (Liu et al., 2025) | 200 | 2 | 🔋 | ✓ | ✓ | ✗ |
| Leaf (An et al., 2023) | 30 | 4 | 🔋 | ✗ | ✗ | ✗ |
| ING (Zhou et al., 2024a) | 56 | 5 | 🔋 | ✗ | ✗ | ✗ |
| ISG (Chen et al., 2024) | 1,150 | 8 | 🔋 | ✓ | ✗ | ✗ |
| IDEA (Liang et al., 2024) | 100 | 10 | 🔋 | ✗ | ✓ | ✗ |
| GDT (Huang et al., 2024b) | 200 | 13 | 🔋 | ✓ | ✓ | ✗ |
| **T2IS-Bench** | 596 | 26 | 🔋 | ✓ | ✓ | ✓ |

matching, clarity, tone, style, depth of field, atmosphere, and artistry of human figures. The final aesthetic score for an image set is computed by averaging the MPS scores of all individual images within the set.

**Holistic Average.** All evaluation metrics are normalized to the range of [0,1], enabling holistic assessment. Considering the significant progress in Aesthetic Quality and Prompt Alignment, we assign different weights when computing the holistic assessment score: 0.2 for Aesthetic Quality, 0.3 for Prompt Alignment, and 0.5 for Visual Consistency. This weighting scheme reflects both the relative importance and the current challenges in T2IS generation.

### B.2 COMPARISON WITH EXISTING BENCHMARKS

As shown in Table 4, compared to previous benchmarks like OpenLeaf, OpenING, and IDEA-Bench, T2IS-Bench with T2IS-Eval offers several key advantages:

**Comprehensive Coverage:** T2IS-Bench provides extensive coverage across 26 diverse categories while maintaining a reasonable benchmark size of 596 tasks. In contrast, existing benchmarks have limited domain coverage, ranging from just 2 domains in Consistory+ to 10 domains in IDEA-Bench.

**Multi-level Evaluation:** Our benchmark enables detailed evaluation at both the individual image level prompt alignment and set-level consistency. While some benchmarks like OpenLeaf and OpenING support sub-image evaluation, and others like IDEA-Bench assess consistency, T2IS-Bench uniquely combines both aspects for comprehensive assessment.

**Adaptive Assessment:** T2IS-Bench provides adaptive evaluation criteria tailored to each instruction's specific requirements. For example, character generation tasks are evaluated based on identity consistency, while process generation tasks focus on logical coherence of steps. This instruction-specific evaluation approach contrasts with existing benchmarks that use fixed evaluation criteria regardless of task type, enabling more nuanced and relevant assessment of model capabilities.

## C DETAILED EXPERIMENTAL SETTINGS

We employ FLUX.1-dev (Labs, 2024) as our base model. We execute a total of 20 denoising steps with a guidance scale of 3.5. Each invidual image has a final resolution of $512 \times 512$ pixels, with initial latent dimensions of $32 \times 32$. For image sets of size 2, 3, and 5, we arrange them in a $1 \times n$ grid layout during the conquer phase. For sets of size 4, we utilize a $2 \times 2$ grid arrangement. For sets larger than 5 images, we adopt a sliding window approach with a window size of 4 and stride of 2.

As analyzed in Appendix G (Figure 8), elongated layouts increase the spatial distance between sub-image latents, which weakens cross-image interactions within the DiT's attention layers. This separation can lead to consistency degradation, such as drifting logos or missing fine-grained details. In contrast, compact layouts maintain latent proximity, allowing for the effective propagation of identity and style cues. Notably, our experiments reveal that the $2 \times 2$ grid configuration achieves superior results, likely due to the closer proximity between regions. In contrast, the $1 \times 5$ arrangement tends to introduce consistency mismatches between the first and last images.

Table 5: Comparison between human, model, and baseline evaluations on visual consistency assessment. We report the Pearson Similarity between model predictions and human annotators (blue: best in each column), as well as qualitative correlation statements.

| Method | Pearson (Bin.) | Pearson (Fine.) | Correlation w/ Human Judgment |
|---|---|---|---|
| CLIP-I | 0.52 | 0.43 | Moderate; mainly global similarity |
| DreamSim | 0.57 | 0.48 | Better detail; unstable across tasks |
| **T2IS-Eval** | 0.78 | 0.61 | Strong agreement with human ratings |
| Human-to-Human | 0.89 | 0.75 | — (Upper bound) |

## D    RELIABILITY ANALYSIS OF T2IS-EVAL

In this section, we conduct experiments to validate the reliability of our T2IS-Eval framework by comparing with two types of human evaluations. Our analysis focuses on assessing how well Qwen2.5-VL's judgments align with human assessments of visual consistency between image pairs.

**Human Binary Assessment.**    We first collected binary judgments (consistent/inconsistent) from human annotators on 200 diverse image pairs. For each pair, three annotators were asked to make a yes/no decision on whether the images were visually consistent. The final human binary label was determined by majority voting.

**Human Fine-grained Assessment.**    We also gathered fine-grained consistency scores from human raters on a 0-5 scale, where 0 indicates complete inconsistency and 5 indicates perfect consistency. This allowed capturing more nuanced degrees of visual consistency that may be missed by binary judgments. Each image pair was rated by three annotators and scores were averaged.

**Alignment between Model and Human Judgments.**    As shown in Table 5, Qwen2.5-VL's binary decisions achieved a Pearson Similarity of 0.78, demonstrating strong alignment with human judgments in distinguishing consistent from inconsistent image pairs. For fine-grained assessment, our model outputs likelihood scores across different consistency levels, offering more nuanced measurements while avoiding potential hallucination issues that could arise from directly generating numerical scores. To evaluate the reliability of this fine-grained assessment, we scaled the model's likelihood outputs from [0,1] to [0,5] to match the human rating scale. This scaled likelihood distributions achieved a Pearson Similarity of 0.61 with human fine-grained ratings, validating that T2IS-Eval can effectively capture subtle variations in visual consistency that align well with human preferences.

## E    ABLATION ON DIFFERENT IMAGE GENERATION BACKBONES

To assess the generality and robustness of AutoT2IS, we conduct extensive ablation studies by applying it across various state-of-the-art text-to-image diffusion backbones, including both open-source (FLUX.1-Dev (13B), Qwen-Image (20B), Hunyuan-Img3.0 (80B)) and closed-source (GPT-4o, NanoBanana, SeedDream4.0) models.

Our results, summarized in Table 6, reveal several key trends: (1) Larger backbones with richer pretraining (e.g., Hunyuan-Img3.0, 80B) naturally achieve higher set-level visual consistency. (2) AutoT2IS consistently yields further improvements across all open-source models, with the most significant absolute gains on the strongest visual backbones. (3) For closed-source commercial models, the Structured Recaption input strategy alone still delivers salient improvements in visual and logical coherence, underscoring the plug-and-play and model-agnostic nature of our approach.

## F    LIMITATIONS AND FUTURE WORK

While our proposed approach demonstrates promising results for generating consistent image sets, there are several limitations and opportunities for future work.

Table 6: Ablation study of AutoT2IS on various backbone models.

| Model | Identity | Style | Logic |
|---|---|---|---|
| *Open-Source Models* | | | |
| FLUX.1-Dev (13B) | 0.359 | 0.414 | 0.356 |
| Qwen-Image (20B) | 0.356 | 0.421 | 0.376 |
| Hunyuan-Img3.0 (80B) | 0.508 | 0.549 | 0.422 |
| *Closed-Source Models* | | | |
| GPT-4o | 0.400 | 0.463 | 0.383 |
| NanoBanana | 0.421 | 0.459 | 0.417 |
| SeedDream4.0 | 0.486 | 0.479 | 0.458 |

**Baseline Nature of Current Approach.** Our current divide-and-conquer strategy, while effective, represents a baseline approach to the problem of image set generation. More sophisticated methods for ensuring consistency and managing the generation process could potentially yield better results.

**Resolution and Memory Constraints.** The current implementation is optimized for generating images at $512\times512$ resolution. When scaling to higher resolutions such as $1024\times1024$, the memory requirements increase substantially, particularly during the conquer phase where multiple images need to be processed simultaneously. This limitation becomes especially pronounced when generating larger image sets, necessitating further research into memory-efficient techniques.

**Consistency Challenges.** Our experiments reveal several key challenges in T2IS generation, including maintaining visual consistency across spatially distant positions, handling complex instructions requiring strong reasoning, and coordinating detailed sub-stories across multiple frames. We provide detailed analysis and examples of these challenges in Appendix G.

**Future Work.** Future research could focus on:

- Developing memory-efficient architectures for high-resolution generation through progressive growing or sparse computation methods (Zhang & Agrawala, 2025) to address the resolution and memory constraints.

- Exploring large-scale pretraining strategies (Google Cloud, 2025; OpenAI, 2025) and instruction understanding to enhance the model's capability in handling instructions requiring strong reasoning and domain knowledge.

- Improving long-range consistency through advanced attention mechanisms and hierarchical modeling to better maintain visual coherence across spatially distant positions.

- Developing better prompt alignment strategies to effectively coordinate detailed sub-stories and ensure narrative coherence across multiple frames.

## G  FAILURE CASES ANALYSIS

We analyze three main types of failure cases observed in our experiments:

**Consistency Degradation in Grid Layout.** In the 1*5 grid layout, we observe consistency degradation across the image sequence. Figure 8 shows this limitation where the logo's appearance noticeably shifts between the first and last images in the sequence, highlighting the challenge of maintaining visual consistency across spatially distant positions.

**Complex Instruction Understanding.** Figure 9 demonstrates the model's limitations in precisely following complex instructions that require strong reasoning and domain knowledge. When given instructions involving physical laws while maintaining specific constraints across the sequence, the model struggles to accurately execute all requirements simultaneously.

**Story Prompt Alignment in Detailed Story Generation.** As shown in Figure 10, while our method maintains overall style and character consistency, coordinating detailed sub-stories across multiple frames remains challenging. The model struggles to perfectly align individual sub-stories and local details when generating rich narratives that must both work independently and contribute to a cohesive sequence. This highlights the need for better prompt alignment strategies in complex storytelling.

These limitations point to important areas for future research, including better handling of long-range consistency and improved instruction understanding for complex scenarios requiring domain knowledge and precise control.

# H    ANALYSIS OF CHALLENGE CATEGORIES

we include below a summary table showing the performance across five high-level categories. As shown, tasks that require strong logical consistency, such as Story and Process generation, pose significant challenges to current models in terms of maintaining logical coherence across the image set. Moreover, Story Generation often involves multiple distinct identities, making identity consistency particularly difficult for existing models. These findings help reveal the specific weaknesses of current approaches and can guide future research.

Table 7: Performance Summary Across High-Level Categories

| Category | Dimension | Show-o | Gemini & Flux-1 | Gemini & SD3 | AutoT2IS |
|---|---|---|---|---|---|
| Character Generation | Style | 0.438 | 0.439 | 0.409 | 0.520 |
| Character Generation | Identity | 0.441 | 0.454 | 0.435 | 0.609 |
| Character Generation | Logic | 0.495 | 0.516 | 0.515 | 0.510 |
| DesignStyle Generation | Style | 0.353 | 0.343 | 0.344 | 0.391 |
| DesignStyle Generation | Identity | 0.285 | 0.285 | 0.302 | 0.339 |
| DesignStyle Generation | Logic | 0.405 | 0.471 | 0.452 | 0.478 |
| Story Generation | Style | 0.196 | 0.235 | 0.187 | 0.405 |
| **Story Generation** | **Identity** | **0.120** | **0.145** | **0.115** | **0.262** |
| **Story Generation** | **Logic** | **0.153** | **0.182** | **0.179** | **0.237** |
| Process Generation | Style | 0.195 | 0.201 | 0.198 | 0.346 |
| Process Generation | Identity | 0.139 | 0.143 | 0.142 | 0.314 |
| **Process Generation** | **Logic** | **0.185** | **0.222** | **0.212** | **0.274** |
| Instruction Generation | Style | 0.265 | 0.310 | 0.311 | 0.437 |
| Instruction Generation | Identity | 0.190 | 0.231 | 0.238 | 0.315 |
| Instruction Generation | Logic | 0.197 | 0.256 | 0.250 | 0.315 |

# I    MORE RESULTS FOR RELIABILITY ANALYSIS

## I.1    STATISTICAL SIGNIFICANCE ANALYSIS

We conducted additional experiments and reported the mean and standard deviation across three runs with different random seeds to analyze the effect of random variation. (Due to computational constraints, we focused on methods with manageable inference costs.)

As shown in Table 8, the mean scores are highly consistent with the single-run results originally reported, and the standard deviations are very small, indicating that the observed improvements are statistically significant. Importantly, the conclusions drawn from the multi-run experiments remain identical to those from the single-run results. Moreover, given the large number of instructions (596 in total), the aggregate scores are inherently robust to small fluctuations caused by random seeds.

In addition to reporting mean ± standard deviation with different random seeds, we have now conducted paired t-tests between some methods over the 596-instruction evaluation set in Table 9.

Table 8: Mean and Standard Deviation of Evaluation Metrics Across Three Runs.

| Method | Identity | Style | Logic |
|---|---|---|---|
| Show-o | 0.239±0.005 | 0.293±0.006 | 0.292±0.005 |
| Janus-Pro | 0.240±0.021 | 0.287±0.020 | 0.320±0.029 |
| Gemini & Flux-1 | 0.253±0.004 | 0.307±0.005 | 0.333±0.004 |
| Gemini & SD3 | 0.260±0.014 | 0.303±0.014 | 0.329±0.008 |
| AutoT2IS (ours) | 0.353±0.008 | 0.412±0.006 | 0.353±0.003 |

Table 9: P-value Matrices for Paired T-tests Across Evaluation Dimensions.

| Method | Show-o | Janus-Pro | Gemini & Flux-1 | Gemini & SD3 | AutoT2IS (ours) |
|---|---|---|---|---|---|
| Show-o | — | 0.5968 | 0.0230 | 0.1785 | 0.0025 |
| Janus-Pro | 0.5968 | — | 0.2510 | 0.3058 | 0.0134 |
| Gemini & Flux-1 | 0.0230 | 0.2510 | — | 0.5456 | 0.0018 |
| Gemini & SD3 | 0.1785 | 0.3058 | 0.5456 | — | 0.0085 |
| AutoT2IS (ours) | 0.0025 | 0.0134 | 0.0018 | 0.0085 | — |

These results clearly indicate that AutoT2IS (ours) achieves statistically significant improvements ($p < 0.05$, often $p < 0.01$) over baselines.

## I.2 LLM Performance and Reliability

**Consistency Evaluator.** In T2IS-Eval, the LLM is utilized solely for generating the initial question criteria. These criteria undergo a manual review process to filter out ambiguities and hallucinations. Once finalized, they are consistently applied across all methods being compared, ensuring fairness and reliability. Our manual verification process indicates that over 90% of the generated criteria are reliable. The remaining cases are either regenerated or manually rewritten to ensure quality.

**Structured Recaption.** For AutoT2IS, we selected DeepSeek-R1 after evaluating several alternatives, as shown in Table 10. DeepSeek-R1 demonstrated superior performance at a significantly lower cost. This module employs an LLM as a proxy for structured prompt generation, which does not affect our overall contribution. It could be substituted with human-written captions, albeit at a higher cost. With advancements in LLM technology, its effectiveness is expected to improve further.

Table 10: Performance of Different Structured Recaption Variants.

| Variants | Aesthetics | Alignment | Consistency |
|---|---|---|---|
| DeepSeek-R1 | 0.520 | 0.742 | 0.376 |
| GPT-4o | 0.514 | 0.735 | 0.360 |
| Gemini 2.0 | 0.515 | 0.724 | 0.362 |
| DeepSeek-V3 | 0.508 | 0.718 | 0.325 |
| GPT-4o-mini | 0.502 | 0.721 | 0.320 |

## J Ablation Studies with Visualizations

Figure 11 illustrates the impact of our key components through visual examples. Without Structured Recaption (SR), the model lacks comprehensive global semantic constraints, which significantly impairs its ability to maintain consistency across the image set. While individual images may appear well-formed in isolation, they exhibit noticeable discrepancies in visual elements and artistic style when viewed as a sequence. This demonstrates that SR plays a crucial role in enforcing coherent visual and semantic relationships throughout the generated image set.

When Set-Aware Generation (SG) is removed, although the images retain some consistency in their overall visual aesthetic, they suffer from noticeable quality degradation and reduced alignment with the original prompts (as evidenced by the portable gaming console example in the first image). Furthermore, maintaining consistent representation of detailed elements across the image set becomes substantially more challenging.

In contrast, our full method successfully combines both components, demonstrating significant improvements over the ablated versions by achieving high-quality, consistent, and well-aligned image sets that overcome the limitations observed in each individual component.

We further analyze the effect of different step sizes in our divide-and-conquer strategy. The step size N determines the granularity of the divide phase, with the remaining steps allocated to the conquer phase. As shown in Figure 11, smaller step sizes (N=1) lead to semantic confusion and loss of details, as evidenced by the portable gaming console example (similar to the case without Set-Aware Generation). The initial semantic imprecision makes it difficult to maintain consistency of detailed elements during the later conquer phase. In contrast, with larger step sizes, individual images tend to maintain their own content, resulting in overall inconsistency across the set. This demonstrates the critical importance of balancing step sizes between the two phases. We found that a ratio of 2:20 provides an optimal balance point. Moreover, this ratio can be generalized - similar balanced results can be achieved with proportional ratios like 3:30 and 4:40.

## K    IMPLEMENTATION DETAILS OF EXISTING METHODS

For evaluating various generation methods, we implement and compare the following approaches:

**Compositional Models.** We use gemini-2.0-flash (Team et al., 2023) to decompose the T2IS instructions into sub-captions, which are then fed into state-of-the-art T2I models including Stable Diffusion 3 (Esser et al., 2024), Pixart (Chen et al., 2023), Hunyuan (Li et al., 2024b), and Flux-1 (Labs, 2024). For each model, we follow their official implementation settings with consistent configurations: number of inference steps and guidance scale as specified in their default scripts, and a unified resolution of 1024*1024 pixels.

**Unified Models.** We evaluate Anole (Chern et al., 2024), Show-o (Xie et al., 2024), and Janus-Pro (Chen et al., 2025) using their official implementations and pretrained weights. Among these models, only Anole demonstrates native capability for multi-image generation, while Show-o and Janus-Pro exhibit limitations in comprehending complete T2IS instructions. Due to their relatively weak instruction understanding abilities, we augment their performance by utilizing the sub-captions approach from Compositional Models.

**Agentic Models.** We evaluate ChatDit (Huang et al., 2024d) and ISG-Agent (Chen et al., 2024) which employ multi-step planning and generation processes. Instructions are directly fed into their official scripts to obtain the generated image sets for evaluation.

**Commercial Models.** We also evaluate commercial models including Gemini 2.0 Flash (Google Cloud, 2025) and GPT-4o (OpenAI, 2025). For Gemini 2.0 Flash, we utilize their latest available APIs which directly support multi-image generation, allowing us to evaluate the returned image sets uniformly. Since GPT-4o can only generate single images, we modify the instructions by prepending "please generate [num]-grid images" to produce multi-grid outputs that we then manually segment. As GPT-4o's API was not publicly available during our testing period, we developed automated scripts to interact with their official web interface for input submission and image collection.

# L    VISUALIZATION GALLERY

This section provides a comprehensive visualization gallery that serves two key purposes: (1) demonstrating the diversity and coverage of T2IS-Bench through carefully curated examples across different categories, and (2) showcasing AutoT2IS's capabilities in handling these diverse text-to-image set generation scenarios. For each category, we present detailed task descriptions, example instructions, and AutoT2IS-generated image sets to illustrate both the benchmark's systematic organization and our method's effectiveness in generating high-quality, consistent image sets.

---

Character Generation: *Multi-Scenario*

```
# Group: Character Generation

# Introduction for this subcategory
This subcategory focuses on generating consistent character
representations across different scenarios while maintaining
visual style and character identity.

# Tasks
Task 1: Generate a set of watercolor illustrations featuring a playful
puppy in different scenarios. The set should include playing in a
grassy yard, swimming in water, wearing a training harness,
chasing a colorful ball, and wearing a cozy sweater.
```

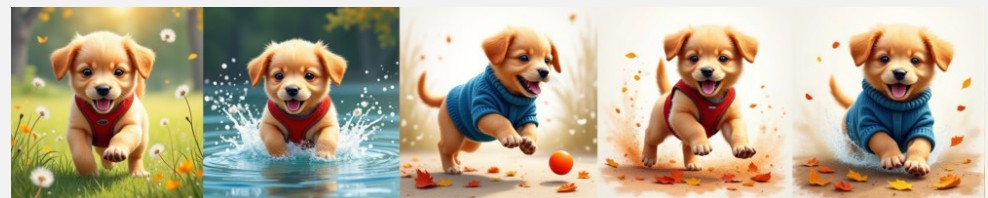

```
Task 2: Create a series of 3D animated images showing a happy hedgehog
in various situations. The set should include resting in a cozy nest,
wearing a miniature jacket, sporting a small collar, dressed in a
festive outfit, and adorned with a flower crown.
```

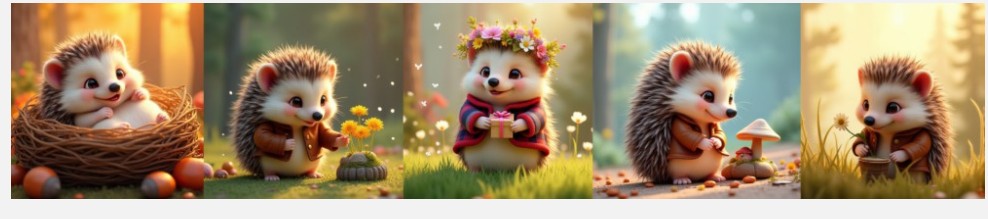

```
......
```

---

Character Generation: *Multi-Expression*

# Group: Character Generation

# Introduction for this subcategory
This subcategory focuses on generating consistent character expressions while maintaining visual style and character identity.

# Tasks
Task 1: Please generate a cowboy gunslinger character in a western comic style, showing only the head or upper body. He wears a wide-brimmed hat, has stubble, and a cigar in his mouth. Generate a set of 3 images with different expressions: the first image shows him grinning confidently; the second image shows him serious and focused, scanning the horizon; the third image shows him angry, ready for a duel. Ensure all facial expressions are diverse, while his hat and attire remain consistent, with the same character ID in every image.

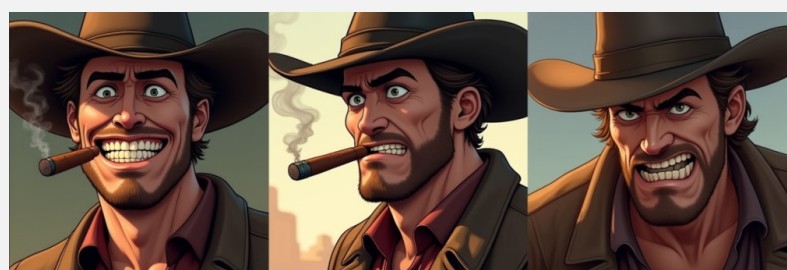

Task 2: Please generate a space explorer character in a futuristic sci-fi style, showing only the head or upper body. He is wearing a sleek space helmet with a transparent visor. Generate a set of 3 images with different expressions: the first image shows him in awe, gazing at something unknown; the second image shows him looking tense, as if detecting danger; the third image shows him determined, ready for an interstellar mission.

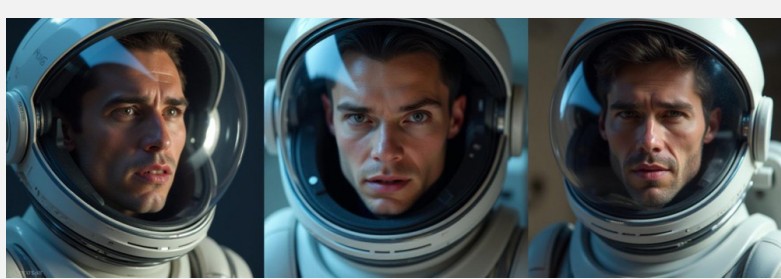

......

Character Generation: *Multi-View*

# Group: **Character Generation**

# Introduction for this subcategory
**This subcategory focuses on generating consistent views of 3D objects and characters from multiple angles while maintaining visual fidelity.**

# Tasks
**Task 1: Please generate four different perspective images of a 3D animated character resembling a fantasy warrior with long, dark hair, pointed ears. The character is dressed in detailed armor with dark and earthy tones, highlighted by silver accents and emblems, suggesting a status of nobility or a special role. His expression is stern , with intense eyes that command attention. The four perspectives are:**
**1. A frontal view that showcases the character's full body...**
**2. A left side profile that captures the silhouette of his face,...**
**3. A rear view that highlights the back design of the armor ...**
**4. A right side profile that provides another angle of his ...**

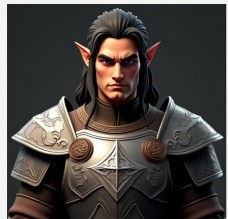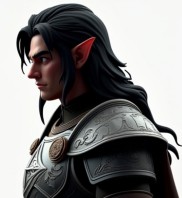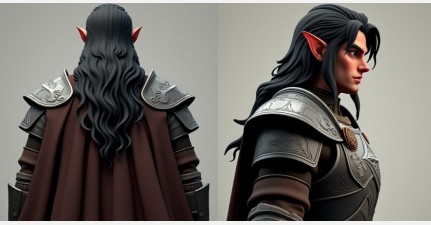

**Task 2: Please generate four different perspective images of a 3D camera model with a classic design. The camera is black with silver accents, featuring a prominent lens and a vintage aesthetic. The four perspectives are:**
**1. A frontal view that showcases the camera's full front,...**
**2. A left side view that captures the camera's profile, ...**
**3. A rear view that emphasizes the back of the camera, ...**
**4. A right side view that provides another angle of the camera, ...**

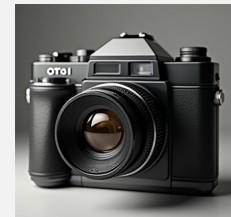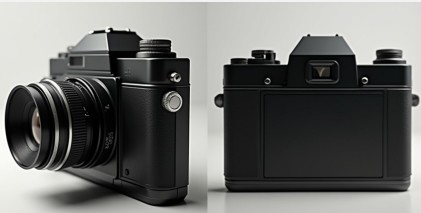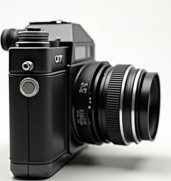

**......**

Character Generation: *Multi-Pose*

# Group: Character Generation

# Introduction for this subcategory
This subcategory focuses on generating consistent character poses
and dynamic movements while maintaining visual style and identity.

# Tasks
Task 1: Please generate a masked ninja character in a monochrome ink
brush style. He wears a traditional shinobi outfit with a katana
strapped to his back. Generate a set of 4 images with different poses:
the first image shows him standing on one foot, arms extended in ....
the second image shows him mid-air, legs spread in a powerful ...
the third image shows him crouching low, gripping his katana hilt ...
the fourth image shows him clinging to a wall, preparing for a ...

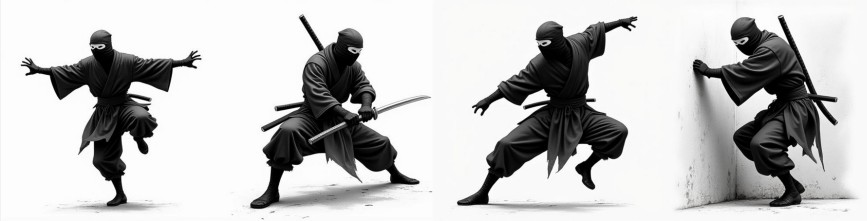

Task 2: Please generate a brave knight character in a realistic style.
He is wearing shiny silver armor, with a determined face, holding ...
the first image shows him standing with both hands holding the ...;
the second image shows him kneeling with one hand gripping the ...;
the third image shows him raising his sword high in an attack ...;
the fourth image shows him swinging his sword to the side in  ...

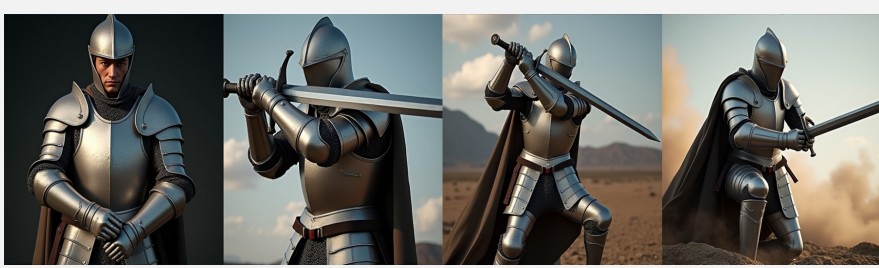

......

Character Generation: *Portrait Design*

# Group: **Character Generation**

# Introduction for this subcategory
This subcategory explores portrait design with emphasis on capturing musicians in different environments and emotional states, focusing on lighting, atmosphere, and storytelling through portraiture.

# Tasks
Task 1: Could you create 4 shots of a street musician with neon violin? First, him playing dramatically under city lights with glowing strings. Second, close-up of tattooed hands on the bow. Third, kids dancing nearby casting colorful shadows. Finally, packing up gear with case open showing stickers. Add lots of urban glow!

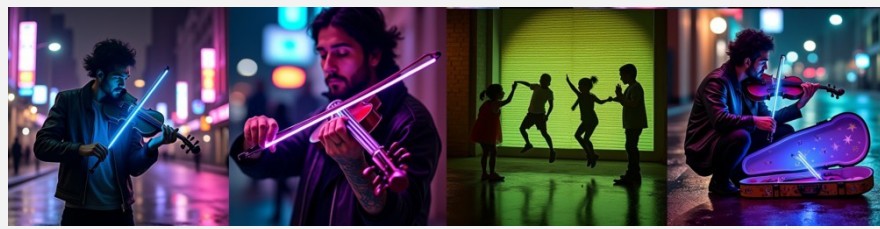

Task 2: Create a series of intimate jazz club portraits capturing a saxophone player. The set should include: an emotional close-up shot showing intense concentration during performance, a natural moment of interaction with band members on stage, and a moody environmental portrait incorporating atmospheric lighting and smoke effects. Focus on conveying the emotional depth and ambiance of the jazz club setting.

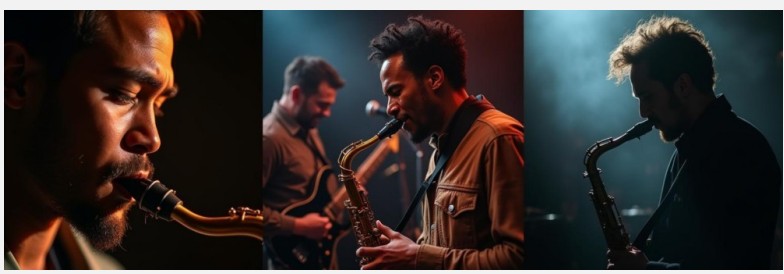

......

Design Style Generation: *Creative Style*

**# Group: Design Style Generation**

**# Introduction for this subcategory**
**This subcategory focuses on maintaining consistent artistic styles across different subjects and scenes while preserving the unique characteristics of each style, from minimalist to complex designs.**

**# Tasks**
**Task 1: Minimalist line style, using soft and smooth black lines to outline the subject. Each painting consists of only a few strokes with large areas of blank space, emphasizing simplicity and elegance. Generate 4 images with the subjects being a crane, a swan, a cat, and a dog. Ensure the line style is consistent across all images, with the subject's contour being simple but recognizable.**

**Task 2: Please generate a set of 5 pixel art-style images ... .**
**the first image shows a morning city street, with vendors ...;**
**the second image is set at noon in the city square, with people ...;**
**the third image shows the entrance to the subway, with a ...;**
**the fourth image depicts a shopping street filled with neon ...;**
**the fifth image shows the city at night, with windows glowing ...;**

**......**

Design Style Generation: *Poster Design*

**# Group: Design Style Generation**

**# Introduction for this subcategory**
This subcategory focuses on creating poster series that maintain consistent visual elements and design principles while highlighting unique focal points.

**# Tasks**
Task 1: Generate a series of 3 posters themed around "Harry Potter and the Prisoner of Azkaban" with a unified, dark, and eerie aesthetic. The color palette is striking, featuring teal, purple, and black. Art Style & Colors: Intense, mystical design with a textured, eerie atmosphere using teal, purple, and black. Background: A dense, dark forest with tall, bare trees, illuminated by a large, bright full moon and wispy clouds. Typography: The movie title "Harry Potter and the Prisoner of Azkaban" in a stylized white font, accompanied by swirling, wave-like teal shapes resembling mist or magical energy. Individual Focus: Poster 1 – Harry's Stand: Focus on Harry Potter at the center of a forest path, with his back turned to the viewer and his wand raised defensively. Emphasize his determined stance against the looming darkness. Poster 2 – The Mysterious Companion: Highlight the figure lying beside Harry (suggesting Sirius Black), adding depth and intrigue to the scene while maintaining the eerie forest ambiance. Poster 3 – The Descent of the Dementors: Emphasize a group of dark, ghostly Dementors with flowing, shadowy forms and purple accents.

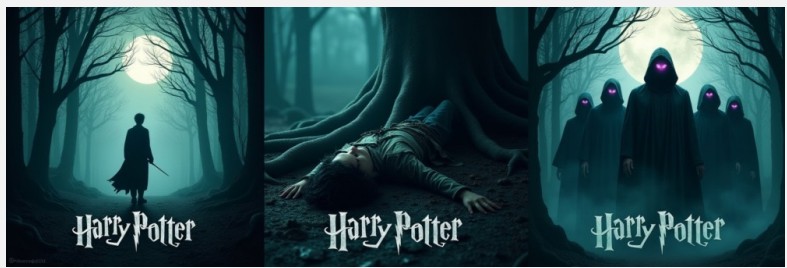

Task 2: Generate a series of 4 vintage racing-themed posters with a unified style. Unified Elements: Header: "PISTON CUP" in large retro black letters (trophy icon for "O") and "RACING SERIES" in small italic text. Background: Stylized palm trees and an orange sky for a California vibe. Individual Focus: Poster 1: Emphasize the dynamic red car "Lightning McQueen" (number "95") with speed lines and a smiling expression. Poster 2: Highlight the turquoise car "DINOC" racing in action. Poster 3: Feature the sleek black car "The King" trailing behind. Poster 4: Combine elements: A "Goodyear" blimp above a checkered flag, A red flame area displaying "LIGHTNING MCQUEEN" in bold yellow, Additional "Cars" characters and logos (CARS, Disney, Pixar) at the bottom.

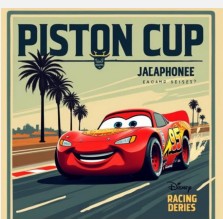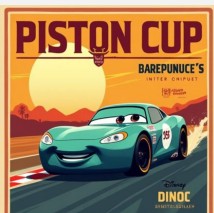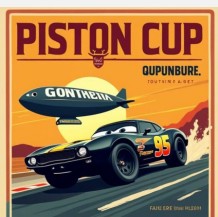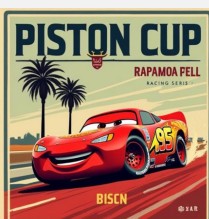

......

Design Style Generation: *Font Design*

# Group: **Design Style Generation**

# Introduction for this subcategory
**This subcategory focuses on applying consistent font designs across
different contexts while maintaining the core typographic style
and visual identity.**

# Tasks
**Task 1: Generate four toy packaging concepts using this playful font:
1 'MAGIC BLOCKS' with floating 3D letters, 2 'ROBOT FRIENDS' with
circuit board textures, 3 'JUNGLE JUMP' wrapped in vine patterns, and
4 'SPACE RACE' featuring planetary orbit formations.
featuring planetary orbit formations.**

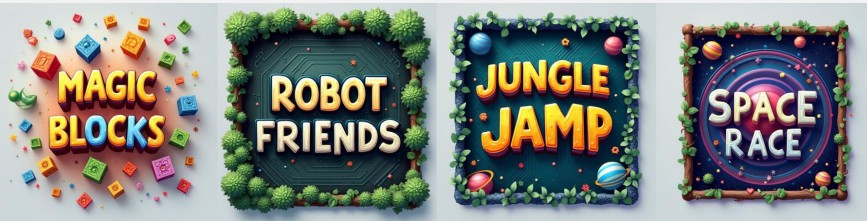

**Task 2: Create four horror movie posters using this bone fracture font
(angular cracks in letters resembling broken limbs). Show: 1 'HAUNTED'
with blood drips on cemetery gates, 2 'SILENT SCREAM' with spiderweb
cracks on abroken mirror, 3 'THE CURSE' glowing in cemetery fog with
splintered edges, 4 'MIDNIGHT' carved into tree bark with glowing
insect trails.**

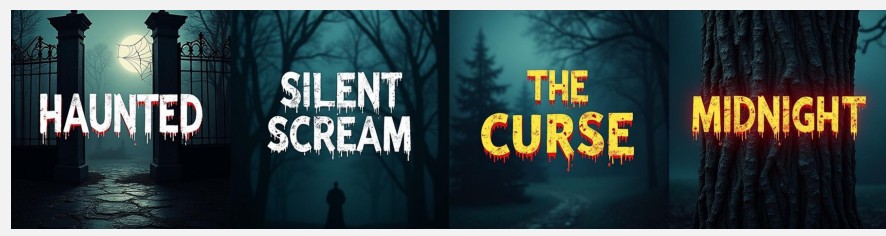

......

Design Style Generation: *IP Product*

# Group: **Design Style Generation**

# Introduction for this subcategory
**This subcategory focuses on maintaining consistent brand identity and design elements across different product applications while preserving visual style.**

# Tasks
**Task 1: Generate product mockups featuring a vintage stamp-inspired logo that embodies old-world travel and exploration, with subtle distressed textures and classic serif typography. Apply the logo on 4 travel-related items: a leather luggage tag, a retro travel journal and a classic travel mug, all rendered in a refined monochromatic palette.**

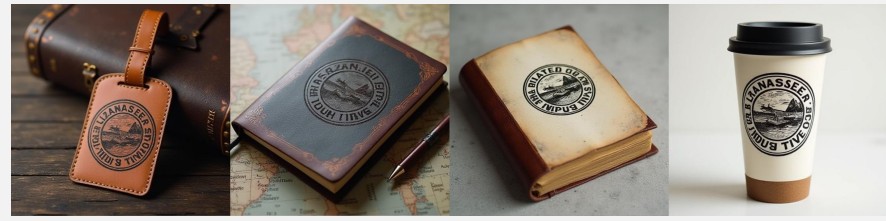

**Task 2: Create product mockups using an organic, hand-drawn logo design that combines minimalist botanical line art with a clean modern typeface, evoking natural purity. Feature this logo on 4 eco-friendly products: a reusable water bottle, an organic cotton tote, a set of bamboo utensils, and a sustainable t-shirt, all presented in a monochromatic style.**

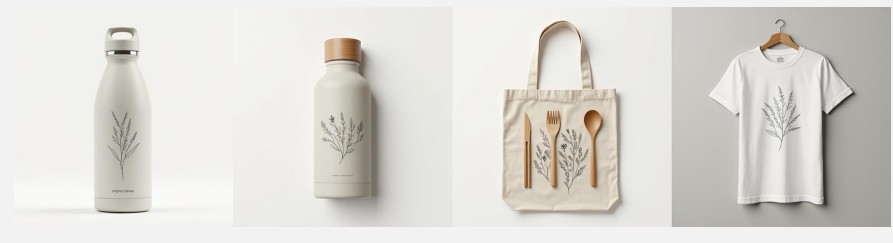

......

Design Style Generation: *Home Decoration*

**# Group: Design Style Generation**
**# Introduction for this subcategory**
**This subcategory focuses on maintaining consistent interior design styles and aesthetic elements across different room perspectives while preserving the overall design theme and atmosphere.**

**# Tasks**
**Task 1: Create four images showcasing a sleek, minimalist living room with neutral tones. Show: 1 An open-concept space with low-profile white sofa, geometric coffee table, and floor-to-ceiling windows showing city views, 2 A monochrome entertainment wall with hidden storage and vertical indoor garden strips, 3 A reading corner with egg-shaped hanging chair and floating shelves holding art books, 4 Close-up of textured beige curtains and matching ceramic floor vases.**

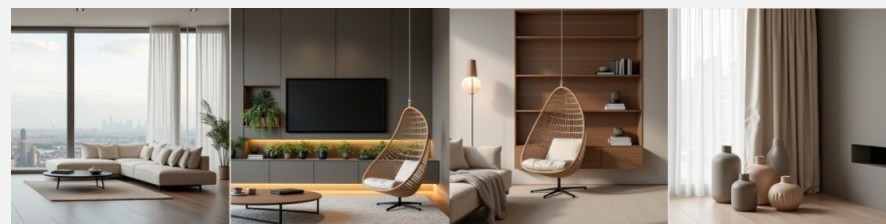

**Task 2: Generate four boho-chic bedroom scenes. Show: 1 A canopy bed with macramé hangings, layered ethnic print blankets, and cushions, 2 Vintage dresser covered in travel souvenirs, crystals, and dried flower arrangements, 3 Window nook with rainbow tassel curtains and a rattan hanging chair, 4 Wall gallery mixing tribal masks, embroidered textiles, and string lights.**

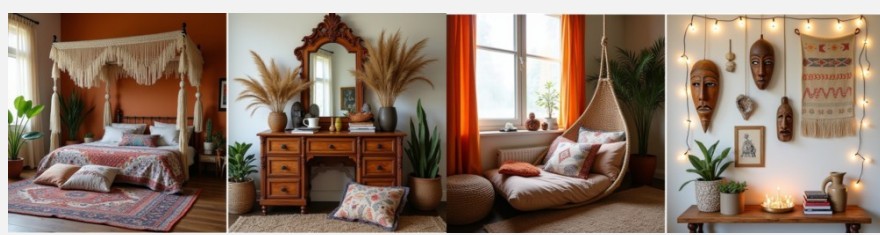

......

Story Generation: *Movie Shot*

# Group: Story Generation

# Introduction for this subcategory
This subcategory focuses on generating consistent cinematic sequences while maintaining visual style, atmosphere, and narrative progression.

# Tasks
Task 1: Develop 3 images for a psychological horror movie. [SCENE-1] Man sits alone in dimly lit room, camera above emphasizes vulnerability, shadows creep in. Use chiaroscuro lighting to create strong contrasts between light and dark. [SCENE-2] He stands abruptly, camera follows as he approaches mirror reflecting something different behind him. Use a subjective camera to show his perspective. [SCENE-3] Final shot shows him from distance in long hallway, doors close as he runs, lighting grows colder and distorted. Use a tracking shot to follow his movement down the hallway. Use lighting and angles to enhance dread.

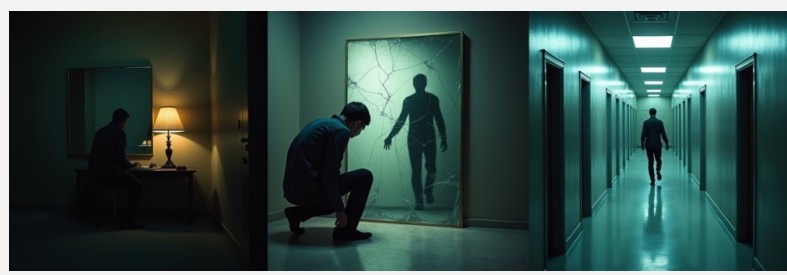

Task 2: Produce 4 images for a musical movie. [SCENE-1] Street performer begins singing, camera captures passionate expression as crowd gathers. Use a medium shot to capture his performance. [SCENE-2] Pull back to show growing diverse crowd united by music. Use a wide shot to show the crowd's reaction. [SCENE-3] Dancers join, colorful costumes contrast urban backdrop. Use a combination of close-ups and wide shots to capture the dance performance. [SCENE-4] Culminate in full production number with dancers, musicians, and singers filling street, dynamic camera movements. Use a circular dolly shot to emphasize the energy of the performance. Use vibrant colors and fluid movements for musical energy.

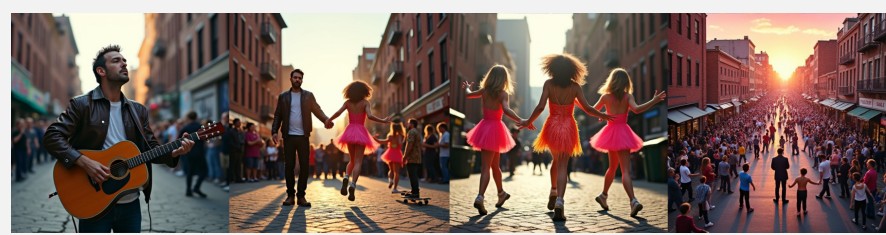

......

Story Generation: *Comic Story*

# Group: Story Generation

# Introduction for this subcategory
This subcategory focuses on generating consistent comic book sequences while maintaining visual style, character design, and story progression.

# Tasks
Task 1: This is a comic book illustration generation task consisting of 5 pages. Title: Intense Superhero Battle – In a sprawling metropolis threatened by chaos and destruction, a courageous red-caped superhero emerges as the city's last hope. Armed with superhuman abilities and unwavering determination, the hero battles against formidable foes to protect innocent civilians and restore peace. Follow the hero's journey through these dynamic comic book panels, capturing the essence of courage, sacrifice, and triumph in the face of overwhelming odds.
 Background Story: The once-bustling city of Metroplis has become a battleground as powerful enemies unleash their destructive forces....

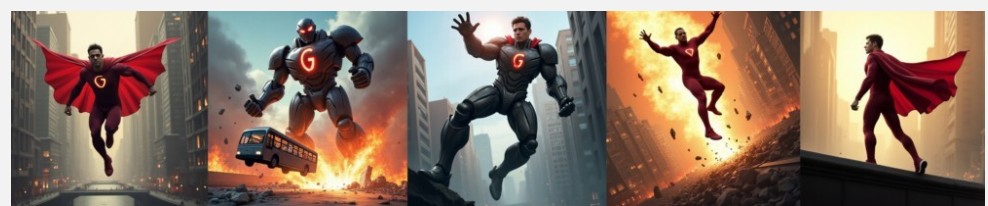

Task 2: This is a comic book illustration generation task consisting of 4 pages. Title: Galactic Warriors: The Space Battle – In the year 3045, humanity has expanded across the cosmos, establishing colonies on distant planets. When a rogue faction threatens this fragile interstellar peace, Captain Lila Voss and her loyal crew aboard the starship Eclipse are dispatched to quell the uprising and restore order to the galaxy.
Background Story: The Eclipse and its crew are part of the United Galactic Federation's elite fleet, renowned for....

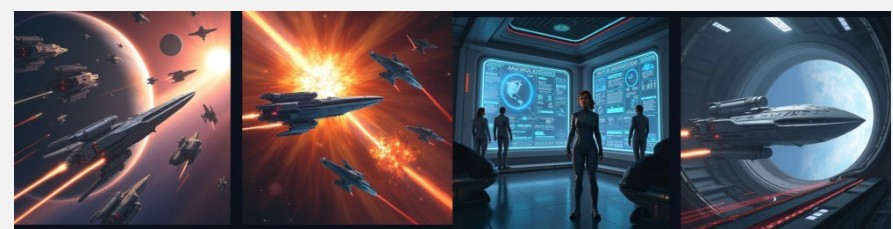

......

Story Generation: *Children Book*

# Group: Story Generation

# Introduction for this subcategory
This subcategory focuses on generating consistent children's picture book illustrations while maintaining visual style, character design, and story progression. Scene and character IDs need to remain consistent throughout the book to ensure stylistic uniformity and character continuity.

# Tasks
Task 1: This is a children's picture book illustration generation task consisting of 4 pages, titled "Little Star's Journey." The main characters include a small star named Twinkle (ID: Twinkle) and the Moon (ID: Moon).
Page 1: Twinkle, a little star, wakes up in the vast night sky and wonders why it shines so far away from the Earth. The Moon gently tells Twinkle that all stars have a journey of their own ...

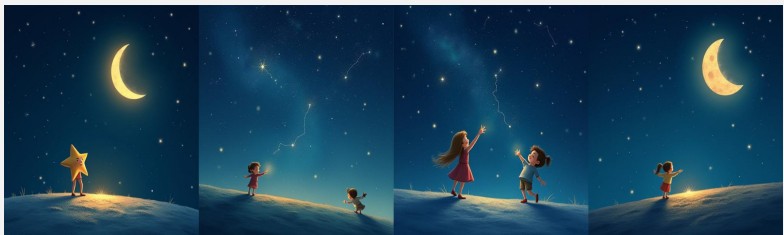

Task 2: This is a children's picture book illustration generation task consisting of 5 pages, titled "The Lion and the Clever Mouse." The characters include a mighty lion named Leo (ID: Leo) and a small but clever mouse named Milo (ID: Milo). Page 1: One sunny afternoon, Leo, the mighty lion, is napping under a tree. Meanwhile, Milo, a little mouse, scurries through the grass and accidentally runs across Leo's paw. The background is a bright, open savanna with tall trees and green grass. Character IDs: Leo, Milo ...

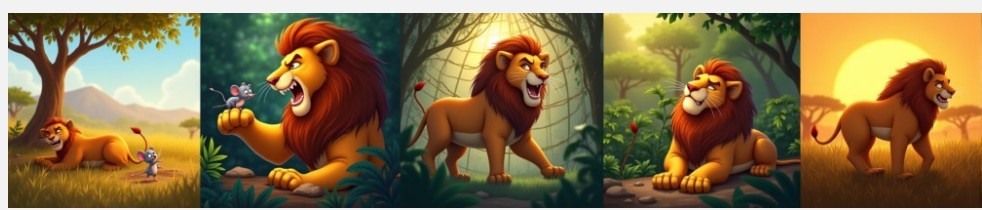

......

Story Generation: *News Illustration*

# Group: Story Generation

# Introduction for this subcategory
This subcategory focuses on generating consistent news illustration series while maintaining visual style and narrative progression. The illustrations should effectively communicate news stories through compelling visuals that complement the written content.

# Tasks
Task 1: This is a news illustration generation task consisting of 3 pages about the revival of a traditional craft in a rural community. Page 1: Show artisans in a workshop practicing the traditional craft, their skilled hands shaping materials with time-honored techniques. Page 2: Depict the process of creating handmade products, from gathering raw materials to the final decorative touches. Page 3: Illustrate the market where these crafts are sold, with customers appreciating the quality and artistry of the unique items. Summary: This set of illustrations celebrates the revival of traditional crafts, providing a visual narrative that complements the news story and engages the audience with its depiction of cultural preservation and artisanal skill.

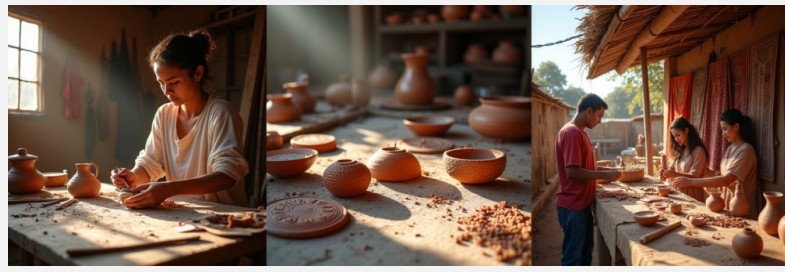

Task 2: This is a news illustration generation task consisting of 3 pages about the construction of a new subway line in a growing city. Page 1: Show the groundbreaking ceremony with officials and community leaders holding shovels of dirt, marking the start of the project. Page 2: Depict underground construction, with tunnel boring machines carving through earth and workers reinforcing the created passages. Page 3: Illustrate the completed subway, with commuters traveling to their destinations and reducing traffic congestion above ground. Summary: This set of illustrations documents the transformation of urban transportation, offering a visual narrative that complements the news story and engages the audience with its depiction of infrastructure development and community progress.

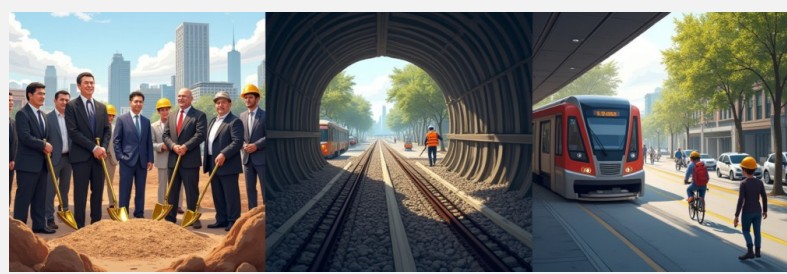

......

Story Generation: *Historical Narrative*

# Group: Story Generation

# Introduction for this subcategory
This subcategory focuses on generating consistent historical narrative illustrations while maintaining visual style and historical accuracy. The illustrations should effectively communicate historical events and their significance through compelling visuals.

# Tasks
Task 1: Generate a set of 5 images depicting key moments of the American Civil Rights Movement from the mid-20th century. The set should include: Rosa Parks refusing to give up her bus seat, showing her determination and the confrontational bus driver; the March on Washington with MLK Jr. delivering his "I Have a Dream" speech at the Lincoln Memorial; Freedom Riders boarding a bus to challenge segregation; protesters during the Selma to Montgomery marches crossing the Edmund Pettus Bridge with police presence; and the signing of the Civil Rights Act of 1964. All images should maintain consistent style while capturing the atmosphere and significance of the Civil Rights Movement.

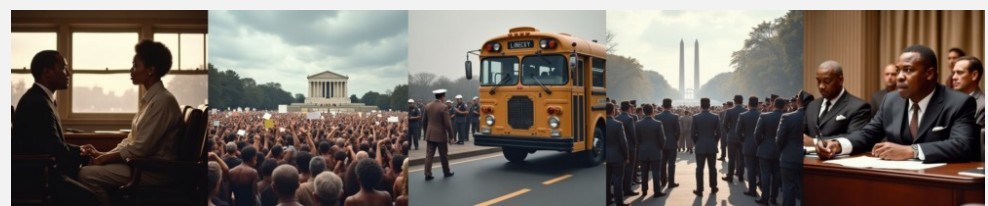

Task 2: Generate a set of images depicting the transformations of Byzantine Constantinople from 330-1453 CE. The set should include: Emperor Justinian overseeing the construction of Hagia Sophia; Varangian Guards defending the Theodosian Walls against Arab siege engines; Fourth Crusade knights looting the Imperial Palace; and Ottoman cannons breaching the walls during the 1453 conquest. All images should maintain a consistent Byzantine mosaic-inspired.

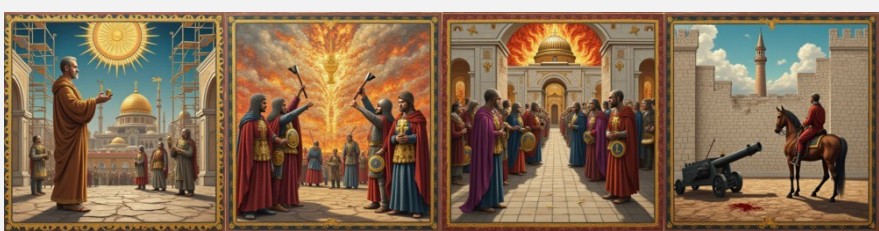

......

Process Generation: *Growth Process*

# Group: `Process Generation`

# Introduction for this subcategory
This subcategory focuses on generating sequential images that depict natural or fantastical growth processes, emphasizing the gradual transformation and development of subjects over time.

# Tasks
Task 1: Please generate a set of images depicting the growth of a beautiful cactus from a tiny sprout to full maturity. The first image clearly shows a small cactus sprout growing in dry desert soil, with vast and vast sand dunes in the background; the second image shows in detail a half-grown cactus with spines beginning to develop naturally, surrounded by sparse desert vegetation and small rocks; the third image shows a fully mature cactus, standing tall and majestic against the backdrop of a stunning sunset over the desert and distant mountains.

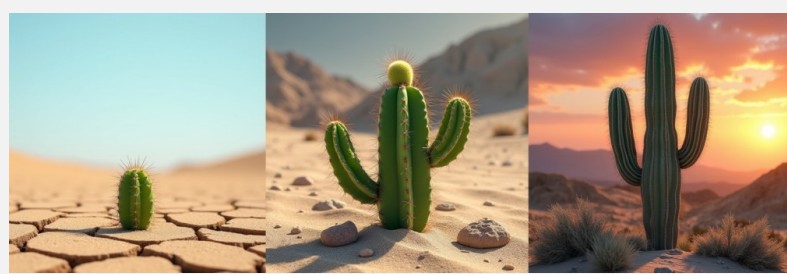

Task 2: Please generate a set of images showing the fascinating transformation of a shapeshifting blob creature from a formless mass to a fully defined entity. The first image features a small, amorphous and mysterious puddle of liquid-like material pulsating on the ground actively. The second image shows the blob beginning to form simple but distinct appendages, experimenting with shape and movement patterns. The third image presents a half-developed creature with distinguishable eyes and features and a flexible, shifting body in constant motion. The final image showcases a fully developed shapeshifter, capable of adopting complex forms and movements, standing confidently and majestically in an alien futuristic city.

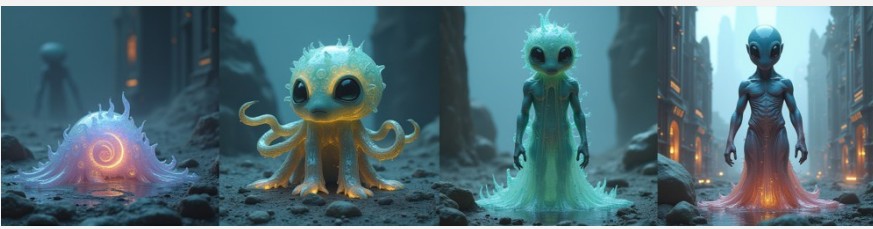

Process Generation: *Draw Process*

# Group: `Process Generation`

# Introduction for this subcategory
This subcategory focuses on generating sequential images that demonstrate the step-by-step process of creating line drawings, highlighting the progression from basic sketches to finished artwork.

# Tasks
Task 1: Please generate a set of images showing the process of drawing a cute, fluffy puppy using simple line art. The first image shows the basic round head sketch with droopy ears and a simple face. The second image adds front legs for a sitting pose. The third image outlines the fluffy body, keeping a soft, chubby look. The final image adds a tiny tail and subtle fur details to complete the drawing.

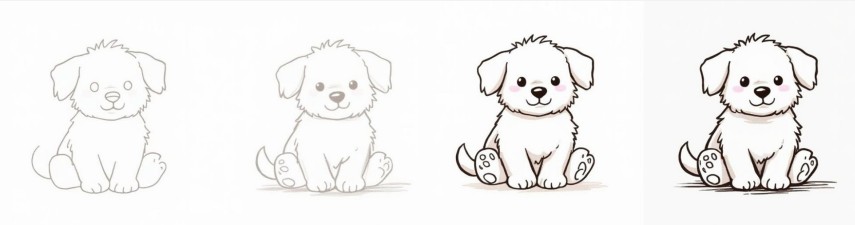

Task 2: Please generate a set of images depicting the process of drawing a charming snowman holding flowers. The first image shows the initial sketch of a round head with a small face and tilted hat adding character. The second image adds the plump body and small arms. The third image includes accessories like flowers in one hand and a small bag on the body. The final image shows the refined drawing with adjusted lines and completed floral arrangement.

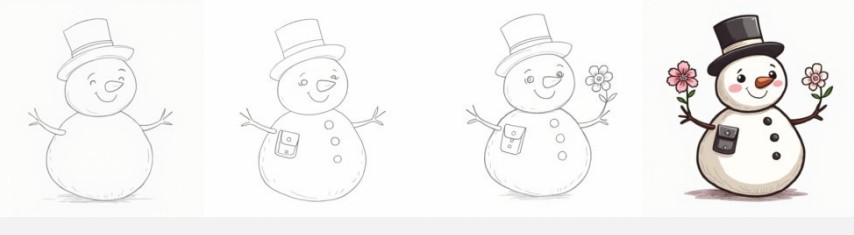

Process Generation: *Cooking Process*

**# Group: Process Generation**

**# Introduction for this subcategory**
This subcategory focuses on generating step-by-step cooking instructions with accompanying images, demonstrating detailed food preparation processes from ingredient preparation to final plating.

**# Tasks**
Task 1: What are 5 steps for cooking pork shoulder steaks on a Weber Kettle Grill, including an image and brief description for each step? Start with preparing the grill for two-zone cooking, then seasoning the meat. Show searing the steaks on the hot side, followed by moving them to indirect heat. Finally, demonstrate checking for doneness and resting the meat before serving.

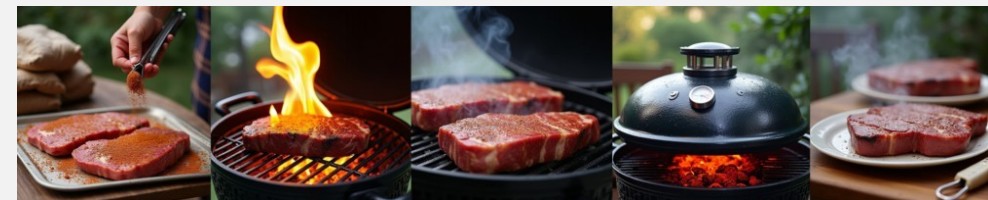

Task 2: Please provide a detailed guide on how to make a bacon sandwich, including 4 key steps. Begin with cooking the bacon until crispy, then toasting the bread slices. Show adding condiments like mayonnaise or lettuce, and finally assembling the sandwich with the crispy bacon and optional toppings like tomato slices.

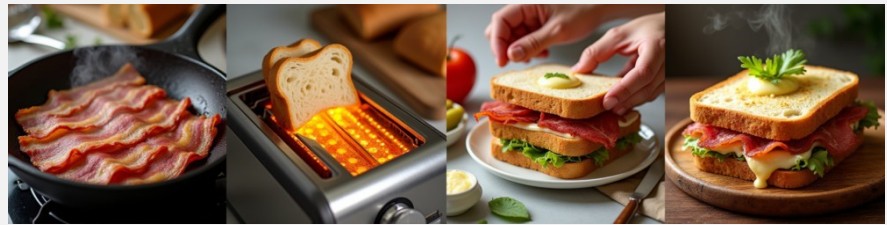

Process Generation: *Physical Law*

```
# Group: Process Generation
# Introduction for this subcategory
This subcategory focuses on generating sequential images that illustrate
natural processes and physical phenomena, demonstrating how these
events unfold over time in accordance with natural laws.

# Tasks
Task 1: Please generate a scene of an apple falling from a tree,
containing 4 images arranged in chronological order, showing
the process of the apple detaching from the tree and
contacting the ground. All images must follow the physical
law of gravity.
```

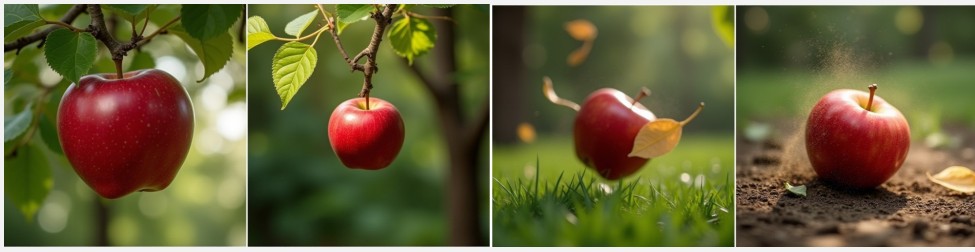

```
Task 2: Generate three images illustrating refraction of light through
a prism. The first image shows a beam of white light
approaching a glass prism. The second image captures the
light entering the prism and bending due to refraction. The
third image presents the light emerging from the prism, split
into a spectrum of colors, demonstrating the dispersion effect.
```

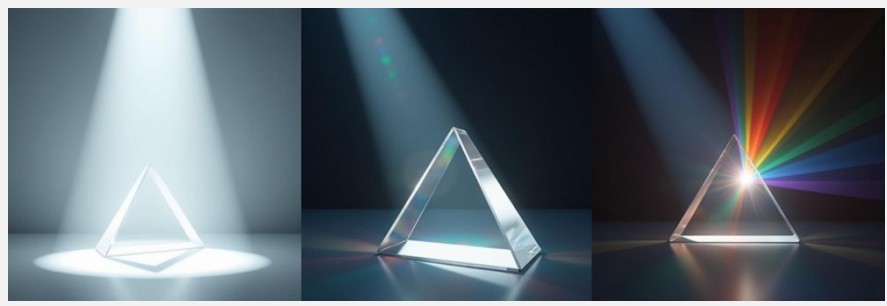

**Process Generation:** *Architecture Building*

**# Group: Process Generation**

**# Introduction for this subcategory**
**This subcategory focuses on generating sequential images that illustrate the complex process of architectural construction, from initial planning to final completion, demonstrating the various stages of building development.**

**# Tasks**
**Task 1: Generate a detailed 4-stage image series illustrating the construction process of a community center.**
**•Stage 1: Community planning and site clearing.**
**•Stage 2: Laying the foundation and constructing the basic framework.**
**•Stage 3: Building multi-use spaces and facilities.**
**•Stage 4: Final interior design, landscaping, and installation of community.**

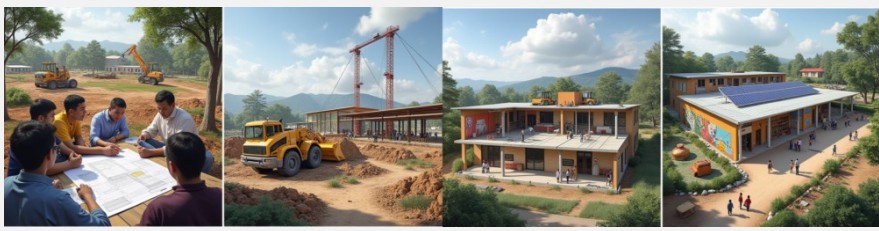

**Task 2: Create a detailed 4-stage image series illustrating the construction process of a high-tech corporate headquarters.**
**•Stage 1: Site clearing and blueprint planning.**
**•Stage 2: Foundation work and structural framework erection.**
**•Stage 3: Installation of smart glass facades and office layouts.**
**•Stage 4: Final finishing touches including landscaping and ....**

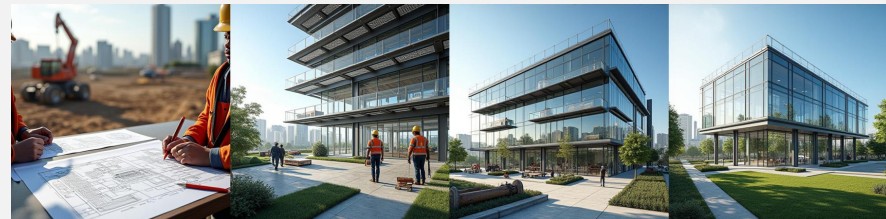

Process Generation: *Evolution Illustration*

# Group: `Process Generation`

# Introduction for this subcategory
This subcategory focuses on illustrating the natural progression of decay and aging processes over time, demonstrating how objects and structures transform through environmental exposure and natural degradation.

# Tasks
Task 1: Generate a scene of a leaf decaying on the forest floor, containing 4 images arranged in chronological order, showing the leaf's progression from a fresh green state to a fully decomposed, brown state. All images must follow the natural laws of decomposition and decay.

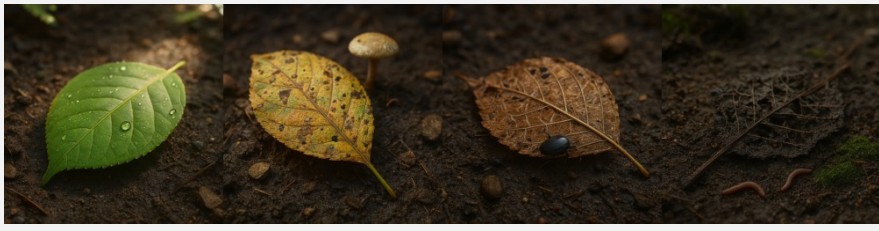

Task 2: Generate a scene of a building aging over time, containing 4 images arranged in chronological order, showing the building transitioning from its newly constructed state to one that shows signs of weathering and decay. All images must follow the physical laws of material degradation and structural stress.

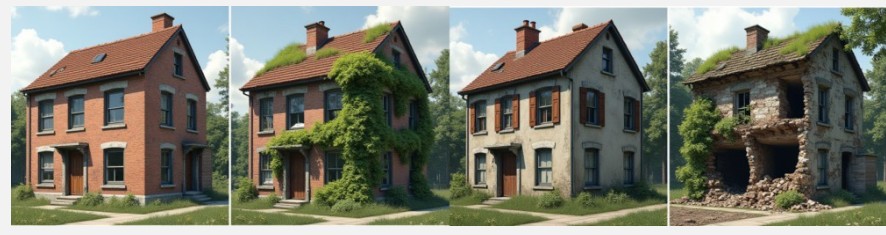

**Instruction Generation:** *Education Illustration*

**# Group:** `Instruction Generation`

**# Introduction for this subcategory**
This subcategory focuses on creating educational content that explains scientific phenomena and natural processes in detail, providing clear and comprehensive illustrations of complex concepts for educational purposes.

**# Tasks**
**Task 1: Generate a detailed scientific explanation of volcanic eruptions, including the underlying geological processes, magma formation, pressure buildup, and the various types of volcanic activities. The explanation should incorporate relevant terminology and physical principles.**

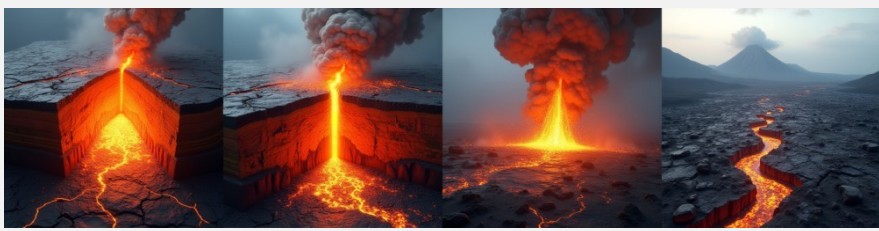

**Task 2: Create a comprehensive explanation of honey formation, detailing the process from nectar collection by bees to the final honey product. Include the biological and chemical principles involved, such as enzyme action, water content reduction, and honey maturation in the hive.**

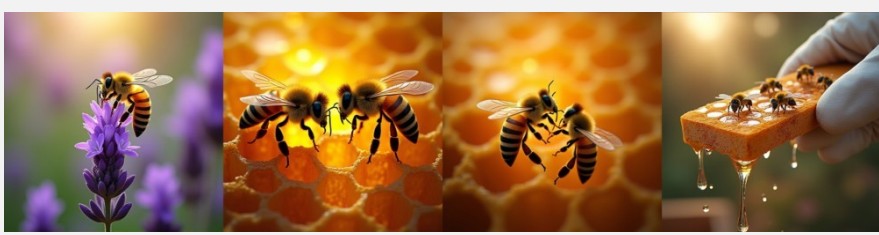

Instruction Generation: *Historical Panel*

**# Group:** Instruction Generation

**# Introduction for this subcategory**
This subcategory focuses on generating detailed historical accounts and analyses of significant cultural landmarks, monuments, and architectural achievements, exploring their historical context and lasting impact on society.

**# Tasks**
Task 1: What is the Terracotta Warriors? Provide a comprehensive history of its construction, cultural significance, and associated events.

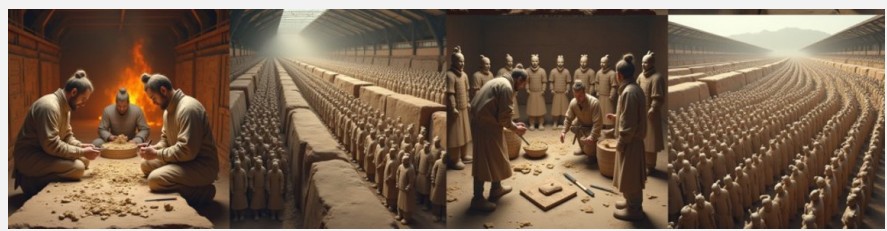

Task 2: Provide a comprehensive overview of the Sydney Opera House, detailing its construction timeline, historical significance, and cultural impact. Additionally, include notable events linked to the venue.

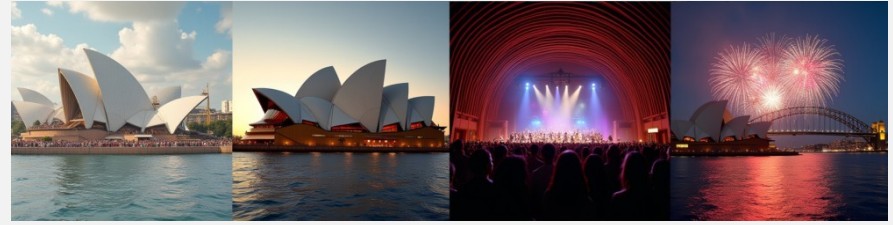

**Instruction Generation:** *Product Panel*

**# Group:** `Instruction Generation`

**# Introduction for this subcategory**
This subcategory focuses on creating clear, step-by-step instructions for various products and DIY projects, ensuring each step is well-illustrated and properly explained for user comprehension.

**# Tasks**
**Task 1: Please provide a detailed guide on the 5 steps of shaving, including an image and brief description for each step.**

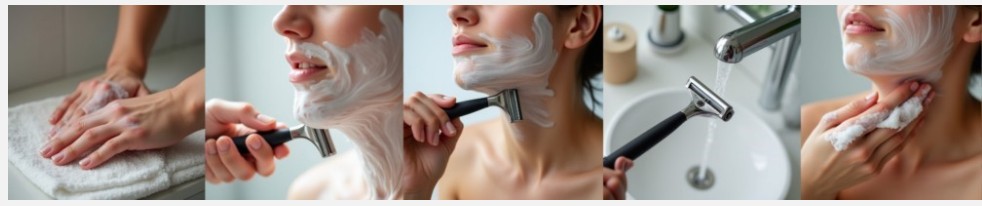

**Task 2: Can you provide a step-by-step guide with images and descriptions for creating DIY hanging rope shelves, specifically 4 steps?**

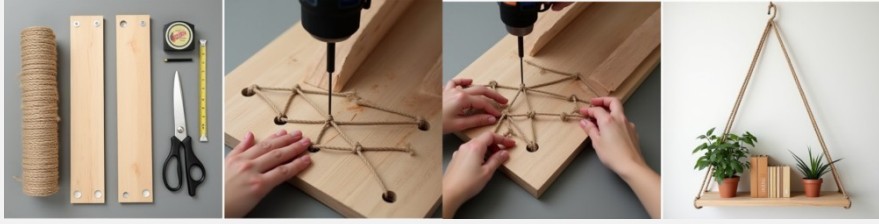

Travel Guide: *Product Panel*

# Group: Travel Guide

# Introduction for this subcategory
This subcategory focuses on creating comprehensive travel guides for popular tourist destinations, highlighting key attractions, cultural landmarks, and providing essential visitor information with high-quality visual content.

# Tasks
Task 1: Create a comprehensive guide to Barcelona's top tourist attractions, including 5 high-quality images of iconic landmarks.

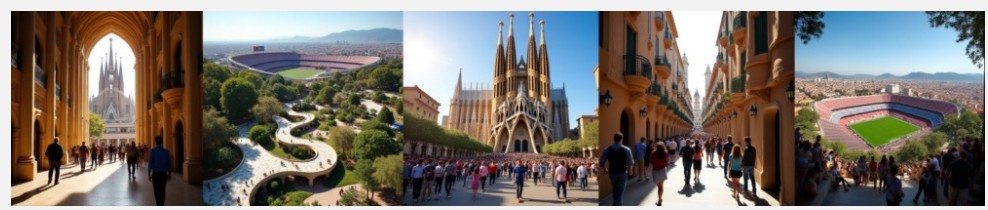

Task 2: Create a comprehensive guide to the top tourist attractions in Rio de Janeiro, including 4 images of its most iconic landmarks.

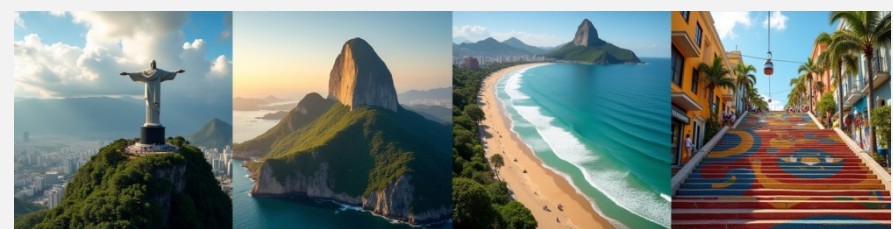

Activity Arrange: *Product Panel*

# Group: Activity Arrange

# Introduction for this subcategory
This subcategory focuses on planning and organizing various events and activities, providing detailed step-by-step guides with visual references to help users create memorable experiences.

# Tasks
Task 1: Please provide a plan for a beach wedding. Generate 4 images showing the setup process and 2 images of the final scene with detailed text descriptions for each step.

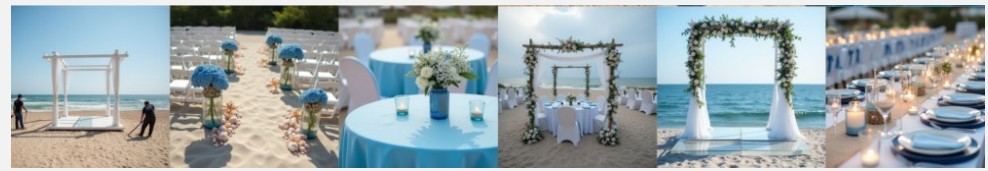

Task 2: Please provide a plan for a Valentine's Day event. Generate 4 images showing the decoration for each step.

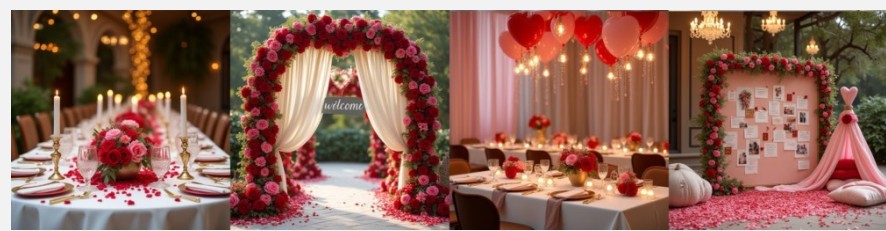

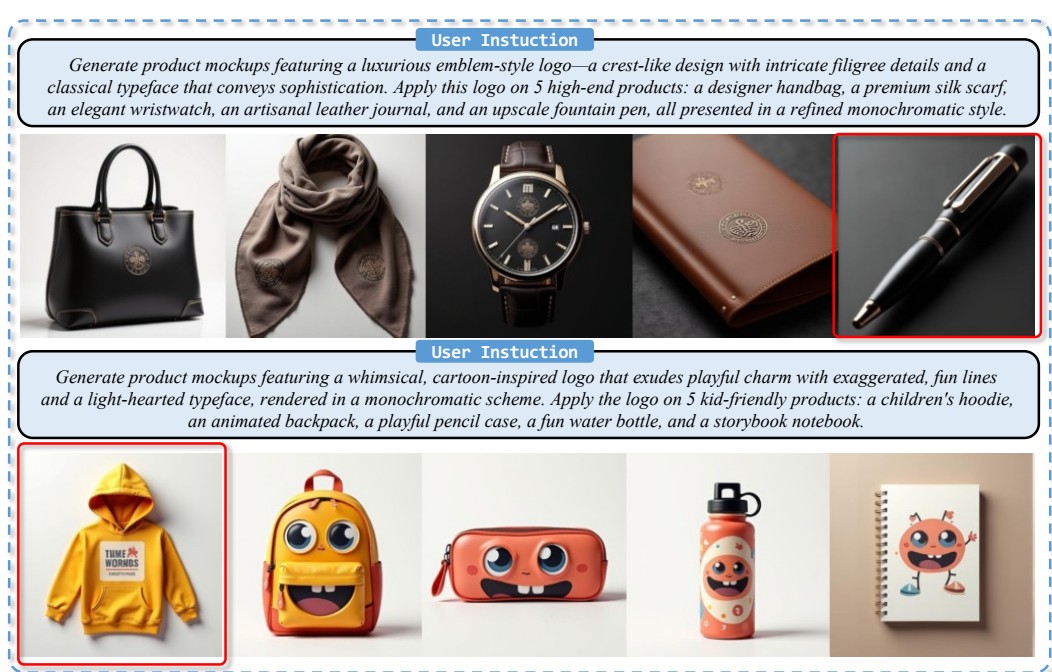

Figure 8: Example of consistency degradation in 1*5 grid layout. Note how the logo's appearance shifts across the sequence, with more pronounced differences between the first and last images.

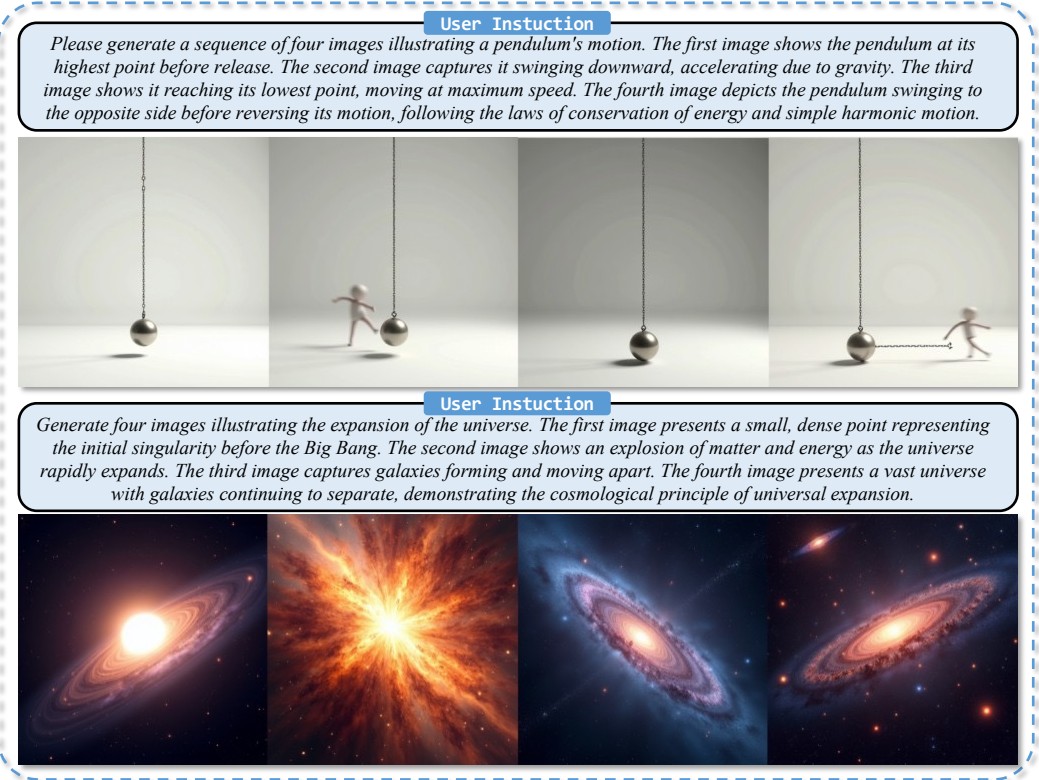

Figure 9: Example of instruction understanding limitations. The model struggles to precisely follow complex instructions requiring strong reasoning and domain knowledge (e.g., physical laws) while maintaining specific constraints across the sequence.

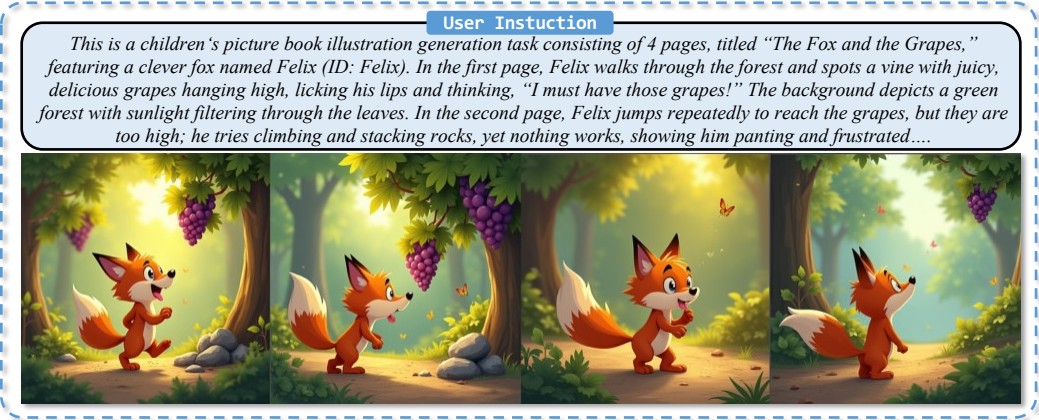

Figure 10: Example of challenges in maintaining story prompt alignment across frames. While our method achieves overall style and character consistency, aligning detailed sub-stories across multiple frames remains challenging.

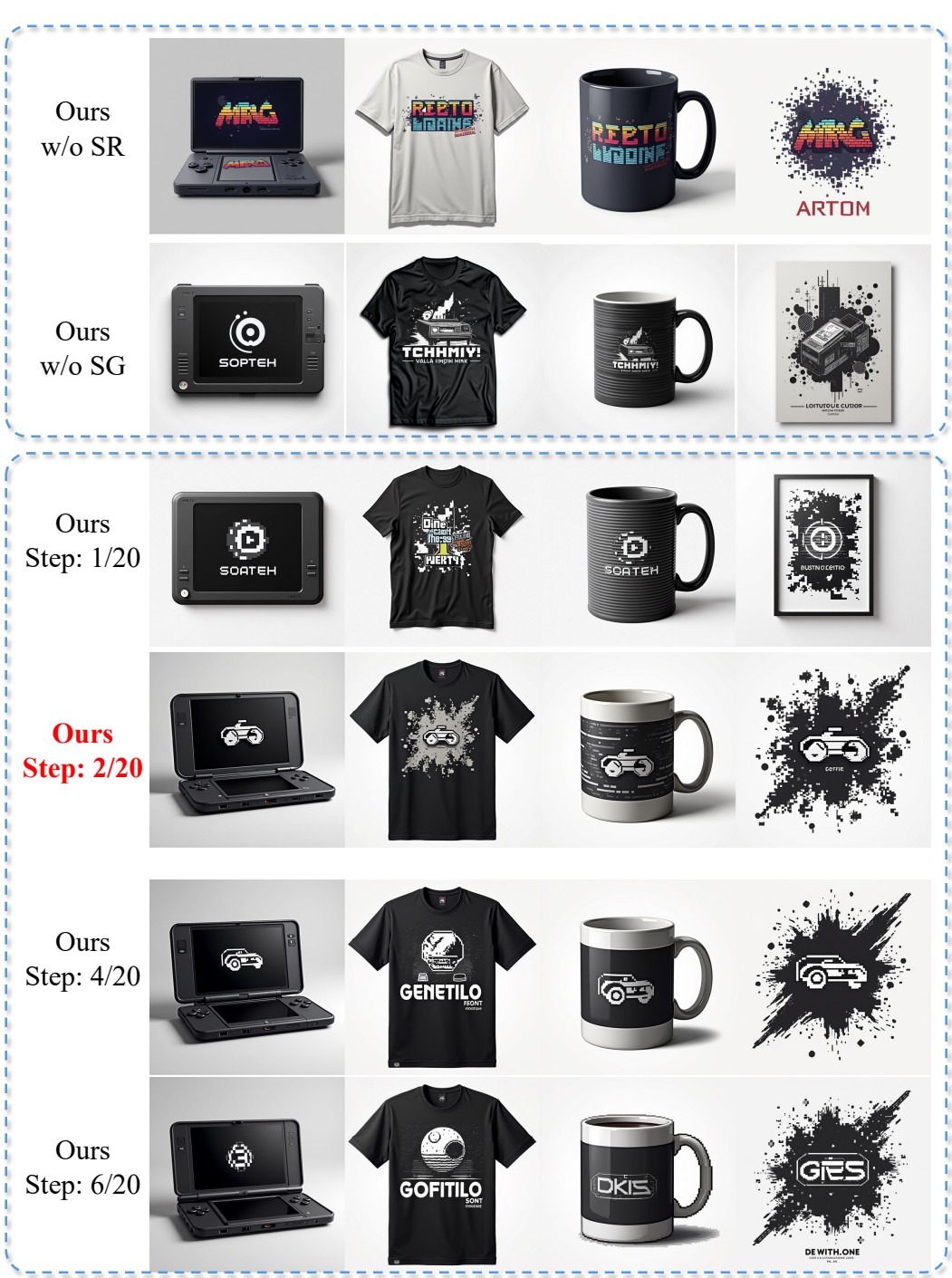

Figure 11: Ablation study visualizing the impact of different components and step sizes. Top row: Results without Structured Recaption (SR) showing inconsistent style and elements. Second row: Results without Set-Aware Generation (SG) showing degraded quality despite some consistency. Third row: Results with step size 1/20 showing semantic confusion. Fourth row: Our full method with optimal step ratio 2/20. Bottom rows: Results with larger step ratios (4/20, 6/20) showing maintained quality but reduced consistency across the set.

