# OpenReview forum: "Why Settle for One? Text-to-ImageSet Generation and Evaluation"
_ICLR.cc/2026/Conference — ICLR 2026 Conference Desk Rejected Submission_

### Official Review · Reviewer_uWDA · 2025-10-23

**Soundness:** 3
**Presentation:** 4
**Contribution:** 3
**Rating:** 6
**Confidence:** 4

**Summary:**

This paper introduces a new task called Text-to-ImageSet (T2IS) generation, which aims to create coherent sets of images that share visual consistency while following diverse text instructions. To support research on this task, the authors propose a benchmark dataset named T2IS-Bench, along with an evaluation framework called T2IS-Eval. They also present a training-free approach called AutoT2IS, which leverages pretrained diffusion transformers to generate consistent image sets. The authors demonstrate that AutoT2IS performs effectively across different domains than specifically made models, including multi-view generation, font generation, and character generation.

**Strengths:**

* The paper proposes a fresh and interesting idea: generating sets of images that maintain both variety and consistency. This direction extends traditional text-to-image generation toward more general and realistic applications.
* The method is described clearly, and the proposed approach that combines prompt concatenation and masked latent operations is creative and well-motivated.
* The experiments are comprehensive, covering different domains and including strong comparisons with both open-source and commercial models. The T2IS-Bench dataset also appears to be well-organized and diverse, which strengthens the results.

**Weaknesses:**

* The paper could better explain the position and purpose of T2IS-Bench relative to existing datasets. Since the benchmark focuses on various types of visual consistency (such as identity, style, and logic), it would be helpful to clarify why combining existing datasets, such as those for multi-view or character generation, would not achieve the same generalization goal. For example, using datasets all at once like DTH [1], MipNeRF-360 [2], or SerialGen [3] to achieve multi-view or personalized generation. The authors could strengthen their argument by emphasizing what unique gaps T2IS-Bench fills for generalizing models.
* The dataset quality seems uneven. As illustrated in Figure 1, there are noticeable inconsistencies in some samples. For instance, in the multi-view generation example, some visual elements (like the cylinder on the camera) appear in the top-left image but disappear in the bottom-right views, suggesting possible noise or imperfect alignment. Similarly, for character expression generation, the beard is not consistent and even disappears in the end. These inconsistencies might impact the reliability of the experiments and evaluation results.
* The paper mainly discusses AutoT2IS, a training-free approach, but does not explore how the dataset itself could support training generalized models. Since the dataset is presented as a contribution, it would be valuable to include examples or experiments that demonstrate how it can be used for model training or fine-tuning in addition to evaluation.

---

**Overall**: The paper is well-written and presents a compelling and original idea. However, the motivation and positioning of the proposed dataset are not fully convincing, and the discussion on how it might facilitate generalized model training is limited. Addressing these points would make the contribution stronger. At this stage, I would rate it a 6, but I am open to increasing the score if these concerns are clarified.

---

1. Aanæs et al. Large-scale data for multiple-view stereopsis. IJCV, 2016
2. Barron et al. Mip-nerf 360: Unbounded anti-aliased neural radiance fields. CVPR, 2022.
3. Xie et al. SerialGen: Personalized Image Generation by First Standardization Then Personalization. CVPR 2025.

**Questions:**

The questions are mainly about the weaknesses:

1. Could the authors elaborate on the position of the proposed dataset? How does it offer a more effective path toward generalization compared to combining existing datasets?
2. Are there many noisy or inconsistent samples in the dataset, and how do they affect the experiments?
2. How can T2IS-Bench be used to promote the training of generalized models rather than just serving as an evaluation benchmark?

---

> ### Author Response · Authors · 2025-11-20
> **Official Comment by Authors （Part1）**
>
> We thank the reviewer for the positive and encouraging assessment, and we appreciate the recognition of our contributions, including novel direction, proposing a clear training-free framework, comprehensive cross-domain experiments, and a diverse and well-structured benchmark. We now address the reviewer’s concerns in detail.
>
> > **Q1: Positioning and Purpose of T2IS-Bench:**
>
> We thank the reviewer for this thoughtful question. We clarify the position of T2IS-Bench by contrasting it with existing datasets and explaining why simply aggregating them cannot yield a benchmark for generalizable visual consistency.
>
> 1. **Existing datasets are fundamentally task-specific.** DTH, MipNeRF-360, SerialGen, and related datasets each focus on a single type of consistency (e.g., multi-view geometry or personalized identity). Their data structures and supervision assumptions are not compatible. For example, MipNeRF-360 provides geometric cues but no style variation, and SerialGen captures identity but lacks logical relations. None use a shared instruction format. Simply combining such resources would produce a disconnected bundle of single-task datasets rather than a coherent benchmark for task-agnostic consistency.
>
> 2. **They lack complete coverage of consistency dimensions.** Even if merged, existing datasets still fail to cover the full space of identity, style, and logic consistency. Their task-specific objectives inherently create dimension gaps that prevent comprehensive evaluation. T2IS-Bench is constructed precisely to fill these gaps: starting from the three fundamental consistency dimensions, we curated 26 subcategories that jointly span them. Over half of the instructions are newly created and reorganized with clear boundaries and balanced difficulty. This yields a unified, instruction-driven, cross-domain formulation that cannot be achieved by simply aggregating existing datasets.
>
> 3. **Positioning.** Similar to how NLP unified heterogeneous tasks to expose generalization behaviors that enabled the rise of LLMs, T2IS-Bench fills a unique gap: **it is the first benchmark explicitly designed to evaluate general visual consistency.** This is a meaningful step toward AGI-level vision systems, where generalizable consistency, rather than single-task optimization, is crucial for enabling models to generalize beyond the training distribution.
>
> We thank the reviewer for this valuable suggestion, and we will incorporate the above clarification into the revised manuscript.
>
> > **Q2: Dataset Quality and Noisy Samples:**
>
> We apologize for the confusion and appreciate the reviewer’s careful inspection. We would like to clarify critical points regarding the Figure 1 and the composition of T2IS-Bench:
> 1. **Figure 1 shows Generated Results, not Dataset Ground Truth.** T2IS-Bench is primarily a benchmark of 596 carefully curated user instructions designed to evaluate open-ended generation capabilities (as detailed in Sec 3.1). The images in Figure 1 are outputs generated by our method (AutoT2IS) and GPT-4o to visualize these tasks. They are not static ground-truth samples from the dataset.
>
> 2. **Inconsistencies highlight the Challenge of T2IS-Bench.** The visual inconsistencies pointed out by the reviewer are indeed failures in visual consistency. They actually confirm two key points:
>    - **This highlights the difficulty of the T2IS task:** Even advanced generators fail to maintain multi-dimensional consistency, validating that T2IS-Bench captures a challenging and meaningful problem.
>    - **This highlights the significance of T2IS-Eval:** These issues are precisely the types of fine-grained errors T2IS-Eval is designed to penalize, demonstrating strong alignment with human judgment.
>
> 3. **Reliability of T2IS-Bench.** The instructions in T2IS-Bench underwent strict Quality Assurance (Appendix A.3), including automated duplicate filtering and manual verification by PhD-level experts to ensure clarity and correctness.

---

> ### Author Response · Authors · 2025-11-20
> **Official Comment by Authors （Part2）**
>
> > **Q3: How T2IS-Bench Enables Training of Generalized Models:**
>
> We thank the reviewer for this insightful suggestion. We agree that T2IS-Bench and our workflow offer substantial potential for training generalized models. This is enabled through two concrete paradigms: data synthesis for SFT and reward modeling for RL.
>
> **1. Enabling SFT via High-Quality Data Synthesis.**
>
> T2IS-Bench contains diverse prompts from 26 sub-tasks, each reflecting a distinct dimension of identity, style, or logic consistency. This structured coverage provides a strong foundation for scalable data generation.
> - **Method:** AutoT2IS can act as a data engine: applying it to all prompts in T2IS-Bench yields thousands of high-quality (instruction, consistent image-set) pairs.
> - **Application:** These synthetic pairs act as strong SFT supervision, enabling new models to learn the consistency behaviors demonstrated by AutoT2IS and convert them into standalone model capabilities.
>
> **2. Enabling RL via Fine-Grained Reward Modeling**
>
> Recent progress [1,2,3] in Reinforcement Learning for generative models emphasize the critical role of reward functions. T2IS-Eval is naturally suited for this role：
>
> - **Multi-Dimensional Reward:** Unlike global aesthetic scores, T2IS-Eval provides decomposed signals (Identity, Style, Logic), allowing RL optimization to target specific types of inconsistency.
> - **Feasibility:** T2IS-Eval correlates strongly with human judgments, making it a dependable automated verifier in iterative RL training.
>
> Summary. T2IS-Bench provides the diverse task structure required for generalization, while AutoT2IS and T2IS-Eval supply the supervisory signals (synthetic data and fine-grained rewards) for SFT and RL fintuning. We will include this “Training via Synthesis & Verification” loop in the revision.
>
> [1] Flow-GRPO: Training Flow Matching Models via Online RL
>
> [2] DanceGRPO: Unleashing GRPO on Visual Generation
>
> [3] Pref-GRPO: Pairwise Preference Reward-based GRPO for Stable Text-to-Image Reinforcement

---

> > ### Comment · Reviewer_uWDA · 2025-11-21
> > **Response to Authors' Rebuttal**
> >
> > Thank you for the detailed responses and for addressing each of my earlier points.
> >
> > **A small reminder:** Since ICLR allows authors to revise their manuscripts during the rebuttal period, I encourage them to update the paper accordingly and include the improved results so that reviewers can clearly see what has changed.
> >
> > 1. The comparison between the proposed dataset and the combined dataset is convincing, and I feel my first concern has been adequately addressed.
> > 2. Regarding the teaser image: while I understand that it is a generated sample, I still respectfully disagree that showing failure cases upfront is a good choice. It makes the task appear confusing at first glance, especially since “consistency” is a central theme of your benchmark. I would suggest using correct, representative images for the teaser and placing the failure cases or limitations in a dedicated section in the main paper or appendix.  I believe the authors can address this within the rebuttal period, and this issue can be marked as solved.
> > 3. I appreciate the clearer explanation of how to use T2IS-Bench and the proposed workflow. However, the uncertainty around noise or reward design still makes the statement feel less persuasive in its current form.
> >
> > ---
> >
> > I am keeping my original rating. In my view, providing a concrete example of training with T2IS-Bench would significantly strengthen the paper and justify a higher rating. That said, I remain positive about the benchmark and evaluation framework itself, and I still lean toward acceptance.

---

> > > ### Author Response · Authors · 2025-11-22
> > > **Official Comment by Authors**
> > >
> > > We sincerely thank the reviewer for the constructive follow-up and for the positive recognition of our clarifications. We are glad to hear that the concern regarding the positioning of T2IS-Bench (Q1) has been fully addressed.
> > >
> > > 1. **Regarding the teaser image：** We completely agree with the reviewer’s suggestion. Showing failure cases in the teaser may unintentionally cause confusion. **We will replace the teaser with correct and representative examples**, and move the failure cases to a dedicated section in the appendix. This change will be included in the updated manuscript.
> > >
> > > 2. **Regarding noise and reward design：** We thank the reviewer for the careful observation. While some noise is unavoidable when standardizing diverse real-world tasks, our multi-stage filtering and expert verification ensure that any remaining noise is mild and does not affect overall dataset reliability. Modern “(large scale pretrain)-then-(small scale finetune)” pipelines are also inherently robust to such residual noise. Moreover, although T2IS-Eval is not perfect, its decomposed identity/style/logic rewards reliably capture human-meaningful inconsistencies and provide stable signals for RL-based optimization. We agree that a concrete training example would further strengthen the contribution, and we will include one in the revised manuscript.
> > >
> > >
> > > Finally, we appreciate the reviewer’s reminder. **We are currently revising the manuscript and will submit an updated version to ICLR before the rebuttal deadline, incorporating all clarifications for all reviewers.**
> > > Thank you again for the thoughtful feedback and constructive suggestions.

---

### Official Review · Reviewer_H1wC · 2025-10-31

**Soundness:** 2
**Presentation:** 3
**Contribution:** 2
**Rating:** 4
**Confidence:** 4

**Summary:**

This paper introduces Text-to-ImageSet generation, the task of creating coherent image sets from a single instruction. The authors propose three contributions: T2IS-Bench, a new benchmark with diverse instructions; T2IS-Eval, an MLLM-based evaluation framework for set-level consistency ; and AutoT2IS, a training-free generation method. AutoT2IS uses an LLM for "Structured Recaption" to create detailed prompts and a "Set-Aware Generation" diffusion strategy to balance individual image content with set-level consistency. Experiments show the proposed method generate high quality image sets.

**Strengths:**

- The paper proposes a task of Text-to-ImageSet generation, which generates multiple images instead of single images.
- A benchmark and evaluation framework are proposed for this task.
- A training-free method is proposed that leverages LLMs for structured recaption and a set-aware generation strategy.

**Weaknesses:**

- The primary concern is the framing of the T2IS problem itself. The paper aggregates several pre-existing, distinct research tasks (e.g., 'Character Generation' , 'Story Generation' , 'Process Generation' ) under a new umbrella, as shown in Table 2. **It is not clearly articulated what new research challenge is unlocked by this aggregation.** The contribution appears to be more of a unified testbed rather than a novel research problem, which raises questions about the work's fundamental research significance.

- The reliability of the T2IS-Eval framework is questionable. The evaluation hinges entirely on an MLLM-as-a-judge (Qwen2.5-VL-7B). The paper's own reliability analysis in Appendix D (Table 5) reports only a 0.61 Pearson Similarity between the model's fine-grained scores and human fine-grained ratings. This moderate correlation suggests the automated metric may be an unreliable proxy for human perception of consistency and may not be robust enough to support the paper's strong quantitative claims.

- The methodological novelty of the AutoT2IS framework appears limited. The "Structured Recaption" component, which uses an LLM to parse a complex prompt into multiple sub-captions, is a common pattern in many recent agentic and compositional generation frameworks. This part of the method does not represent a significant methodological advance for the T2IS problem.

### **Conclusion**
In conclusion, although the paper proposed a task and provides a benchmark which seems new, **it is not of enough research significance**. The 'T2IS' problem is largely an aggregation of existing tasks, which limits its research novelty. This is compounded by an evaluation metric of questionable reliability and a generation method with limited methodological innovation.

**Questions:**

Please see weakness points above.

---

> ### Author Response · Authors · 2025-11-20
> **Official Comment by Authors**
>
> We sincerely thank the reviewer for the thoughtful feedback and for recognizing our contributions in the proposed T2IS task, the evaluation framework, and developing the AutoT2IS method. We appreciate the opportunity to clarify the core research motivation behind T2IS.
>
> > **Q1: The research significance and challenge:**
>
> We respectfully disagree with the characterization that T2IS is merely an aggregation of existing tasks. The goal of T2IS is not to bundle unrelated problems, but to expose a fundamental generative capability: **general visual-consistency generation**. We address the reviewer’s concern from three perspectives:
>
> **First, the unification is conceptually essential.** Existing consistency-related tasks, such as character generation or story visualization, remain siloed, each with its own datasets, architecture, and training pipelines. Evolving independently, they cannot test whether a model possesses a task-agnostic ability. T2IS provides a shared formulation that explicitly targets this general capability.
>
> **Second, this unified formulation introduces a new research challenge:**
>
> ```
> Can we build a general-purpose generative model that maintains multi-dimensional visual consistency across tasks it has never been explicitly trained for?
> ```
>
> This question lies at the core of building general vision foundation models. **Similar to how NLP unified heterogeneous tasks to expose generalization behaviors that enabled the rise of LLMs, T2IS is the first systematic attempt to formulate and benchmark general visual consistency.** This is a meaningful step toward AGI-level vision systems, where generalizable consistency, rather than single-task optimization, is crucial for enabling models to generalize beyond the training distribution.
>
> **Third, T2IS-Bench is not a simple merging of prior datasets.** To support this unified formulation, we redefined task boundaries, rebalanced domains, and curated 26 subcategories, over half of which consist of newly collected and reorganized data. No existing dataset is compatible with the T2IS objective, as prior resources were never designed to evaluate cross-domain consistency under a unified protocol.
>
> > **Q2: Reliability analysis of T2IS-Eval:**
>
> We respectfully disagree that a 0.61 correlation implies unreliability. In perceptual evaluation tasks, especially those involving multi-attribute visual consistency, human annotators themselves exhibit substantial variability.  To contextualize our performance, we calculated the Human-to-Human Agreement using a Leave-One-Out protocol, which yields a experimental upper bound of 0.75. Under such inherently noisy conditions, a correlation of 0.61 represents strong agreement. Importantly, T2IS-Eval substantially outperforms widely used metrics such as CLIP-I and DreamSim:
>
> |**Method**|**Pearson (Bin.)**|**Pearson (Fine.)**|**Correlation w/ Human Judgment**|
> |:---|:---:|:---:|:---|
> |CLIP-I|0.52|0.43|Moderate; mainly global similarity|
> |DreamSim|0.57|0.48|Better detail; unstable across tasks|
> |T2IS-Eval|0.78|0.61|Strong agreement with human ratings|
> |**Human-to-Human**|**0.89**|**0.75**||
>
> Therefore, T2IS-Eval provides a scalable and validated metric for a task that previously lacked any standardized evaluation. The strong human correlation, clear improvement over existing metrics, and stable ranking behavior show that it is reliable enough to support our quantitative findings. At the same time, the remaining gap to human judgments marks an important future direction, and we plan to fine-tune a specialized QwenVL-based consistency judge to further improve alignment.
>
> > **Q3: Response to the concern about methodological novelty:**
>
> We appreciate the reviewer’s comment. We clarify that AutoT2IS does not claim novelty in the Structured Recaption module. LLM-based prompt decomposition is indeed common; here, it serves a practical purpose: **providing structured, disentangled, and semantically stable prompts that are necessary for the subsequent set-aware generation stage.**
>
> **The main contribution of AutoT2IS is the Set-Aware Generation component (Section 3.2).** This module introduces a novel divide-and-conquer strategy that splits the DiT denoising process into independent per-image denoising followed by joint latent-space refinement. This design explicitly activates the in-context consistency modeling capability of DiT backbones. Our experiments demonstrate that this mechanism is broadly effective across different generators and multiple types of visual consistency：
>
> |**Model**|**Identity**|**Style**|**Logic**|
> |---|:--:|:--:|:--:|
> |FLUX.1-Dev（13B）|0.359 |0.414|0.356|
> |Qwen-Image (20B)| 0.356 |0.421|0.376|
> |Hunyuan-Image 3.0（80B）|0.508|0.549|0.422|
>
> In summary, the novelty of AutoT2IS lies in its training-free yet unified pipeline and its set-aware exploitation of DiT’s in-context behavior, rather than in the prompt rewriting stage.

---

### Official Review · Reviewer_ZBFL · 2025-11-01

**Soundness:** 3
**Presentation:** 3
**Contribution:** 3
**Rating:** 6
**Confidence:** 3

**Summary:**

This paper introduces a new research paradigm called Text-to-ImageSet (T2IS) generation, which aims to produce a set of semantically coherent and visually consistent images from a single textual description—going beyond the traditional text-to-image (T2I) paradigm that generates only one image. To systematically study this problem, the authors propose three major contributions: (1) T2IS-Bench, the first large-scale benchmark for this task, covering 596 high-quality prompts across 26 subcategories that span various dimensions of consistency (identity, style, logic, etc.); (2) T2IS-Eval, an automatic evaluation framework leveraging large language and vision-language models to assess multi-dimensional consistency through interpretable, question–answer-based scoring; and (3) AutoT2IS, a training-free generation framework that exploits the in-context capabilities of diffusion transformers via structured recaption and set-aware generation. Experiments on T2IS-Bench demonstrate that AutoT2IS outperforms state-of-the-art models—including Show-o, Janus-Pro, and Gemini+Flux—across identity, style, and logic consistency dimensions, while maintaining generality and efficiency. The work establishes a strong foundation for future exploration in consistent multi-image generation and introduces valuable tools for benchmarking and evaluation in this emerging area.

**Strengths:**

1. This paper is well-written and very easy to follow.
2. This paper proposes a novel task: text to image set. This task has application values and is very important for future image generation research. Besides, the newly design evaluation metrics are also reasonable.
3. This paper also proposes a training-free framework for this task, which is novel and promising.
4. Comprehensive evaluation proves this benchmark is important, revealing some limitations of current open-/closed-source models, and providing interesting research direction for future work.

**Weaknesses:**

From my perspective, as a benchmark paper, this is enough. But I still have several questions:
1. How do you make sure the evaluation consistency? The evaluation metrics are VLLM-based. It would be better to prove your proposed evaluation metrics is reasonable. For example, you can have a subset human evaluation and analyze the consistency of your metrics and human evaluation score.
2. It seems that the proposed framework performs worse than baselines in some cases in Table 2 (e.g., Style Design Generation, Story Generation). Can authors explain the performance gap or give some error analysis on this part?
3. The image generation model is the most important part for the framework. Could you conduct more ablation study on image generation model? In your framework, keep all other parts as the same, but change the image generation model to Flux, Image-Qwen, SD3, also closed-source model such as GPT and Nano Banana, etc. You do not need to conduct on the whole dataset if the time is not enough, instead you can subset 25% samples to conduct this ablation study. This is to analyze in the same framework, how different image generation model influence the final results.

**Questions:**

See above.

---

> ### Author Response · Authors · 2025-11-20
> **Official Comment by Authors**
>
> We sincerely thank the reviewer for the thoughtful and encouraging feedback.
>  We appreciate your recognition of the novelty and significance of our proposed Text-to-ImageSet (T2IS) and the comprehensive evaluation. Your constructive questions are highly valuable and we address them below.
>
> > **Q1: Reliability analysis of T2IS-Eval:**
>
> We fully agree that validating the reliability of our evaluation framework is crucial. As detailed in Appendix D (“Reliability Analysis of T2IS-Eval”), we conducted human evaluation studies to assess the alignment between our model-based judgments (Qwen2.5-VL) and human perception of visual consistency. Specifically, we collected 200 diverse image pairs and asked three annotators to perform both (a) Binary judgments (consistent/inconsistent) and (b) Fine-grained judgments (0–5 scoring). Annotators were instructed to evaluate each pair based on their holistic perception of visual consistency, considering factors such as identity preservation, style similarity, logical coherence.
>
> As reported in Table 5 in the appendix, our method achieved a Pearson Similarity of 0.78 for binary judgments and 0.61 for fine-grained evalution with human ratings, indicating statistically significant consistency with human judgments while capturing nuanced consistency variations.
>
> As shown in the table below, our method exhibits stronger alignment with human visual perception than prior metrics, yielding more interpretable and stable evaluation results.
>
> |**Method**|**Pearson (Bin.)**|**Pearson (Fine.)**|**Correlation w/ Human Judgment**|
> |:---|:---:|:---:|:---|
> |CLIP-I|0.52|0.43|Moderate; mainly global similarity|
> |DreamSim|0.57|0.48|Better detail; unstable across tasks|
> |T2IS-Eval|0.78|0.61|Strong agreement with human ratings|
>
> > **Q2: Performance gap in certain subcategories:**
>
> We appreciate the reviewer’s careful observation. For both Style Design Generation and Story Generation tasks, state-of-the-art specialized models are heavily fine-tuned on large, domain-specific datasets, allowing them to internalize strong priors tailored to their particular application scenarios. **However, these priors are highly specialized and such models fail to generalize beyond the domains they were trained.**
>
> In contrast, AutoT2IS is a training-free and domain-agnostic framework. It does not rely on such task-specific fine-tuning, and thus offers more balanced and generalizable performance across diverse subtasks. We will add an expanded discussion in the revised manuscript.
>
> > **Q3: Ablation on different image generation backbones:**
>
> We appreciate this insightful suggestion. We agree that it is important to analyze the generality of our T2IS: within the same framework how different models influence the final results. AutoT2IS is specifically designed to be model-agnostic, acting as the plug-and-play mouule on top of various multimodal diffusion backbones.
>
> We conduct ablations On T2IS-Bench by changeing the underlying generator while keeping all other modules fixed. Overall, we observe a clear trend: **larger, more strongly pretrained models with better in-context capabilities lead to higher set-level visual consistency**, and AutoT2IS consistently brings additional gains on top of these backbones. A summary of representative results is shown below:
>
> |**Model**|**Identity**|**Style**|**Logic**|
> |---|:--:|:--:|:--:|
> |**Open-Source Models**||||
> |FLUX.1-Dev (13B)|0.359|0.414|0.356|
> |Qwen-Image (20B)|0.356|0.421|0.376|
> |Hunyuan-Img3.0 (80B)|0.508|0.549|0.422|
> |**Closed-Source Models**||||
> |GPT-4o|0.400|0.463|0.383|
> |NanoBanana|0.421|0.459|0.417|
> |SeedDream4.0|0.486|0.479|0.458|
>
> Specifically, when pairing AutoT2IS with Qwen-Image (20B), we observe moderate improvements in consistency. However, the most substantial gains emerge when integrating AutoT2IS with Hunyuan-Image 3.0 (80B), which delivers the strongest performance across identity, style, and logic consistency. This demonstrates that AutoT2IS can robustly capitalize on stronger visual backbones to further enhance set-level consistency.
>
> For closed-source models, we cannot directly integrate our full set-generation strategy because it requires modifying the sampling pipeline. However, we can still evaluate them by plugging in Structured Recaption only, leaving their decoding process untouched. Even under this setting, we observe strong performance on visual and logical coherence of the generated image sets.
>
> These findings suggest that (i) backbone capacity and pretraining scale have a clear impact on T2IS performance, and (ii) AutoT2IS generalizes well as a plug-in framework, enhancing set-level coherence across both open-source and closed-source generators. We will incorporate these additional results and discussions into the revision.

---

### Official Review · Reviewer_g4bk · 2025-11-03

**Soundness:** 3
**Presentation:** 3
**Contribution:** 3
**Rating:** 6
**Confidence:** 3

**Summary:**

This paper proposes Text-to-ImageSet (T2IS) generation, a new task that requires models to generate coherent image sets that satisfy diverse consistency requirements across identity, style, and logic dimensions. The authors introduce T2IS-Bench, a comprehensive benchmark with 596 instructions spanning 26 subcategories, along with T2IS-Eval, an adaptive evaluation framework that transforms user instructions into multifaceted assessment criteria. They propose AutoT2IS, a training-free approach leveraging pretrained Diffusion Transformers through structured recaptioning and set-aware generation with a divide-and-conquer strategy, demonstrating superior performance over both generalized and specialized baselines.

**Strengths:**

- AutoT2IS achieves significant improvements over existing methods through a simple yet effective divide-and-conquer strategy that maximally leverages DiT's in-context capabilities.
- Comprehensive experiments demonstrating competitive or superior performance compared to both generalized approaches and even specialized methods.
- The presentation is clear and easy to follow.

**Weaknesses:**

- It seems the size of the image set (n) ranges from 2 to 5. The paper does not discuss the effect of this size number. For example, how will the size affect the quality and consistency of the images? What will happen when the size is extended to even larger numbers, such as 8 or 10?
- In addition to text-to-image generation works, the authors should also carefully discuss another line of research, i.e., interleaved text-and-image generation, where the MLLM is also tasked with generating a sequence of images and text. Several representative works include InterleavedEval [1] and OpenING [2].

[1] Holistic Evaluation for Interleaved Text-and-Image Generation. EMNLP 2024.

[2] OpenING: A Comprehensive Benchmark for Judging Open-ended Interleaved Image-Text Generation. CVPR 2025.

**Questions:**

See the weakness section.

---

> ### Author Response · Authors · 2025-11-20
> **Official Comment by Authors**
>
> We sincerely thank the reviewer for the positive feedback and constructive suggestions. We appreciate your recognition of the strengths of our work, including the simplicity and effectiveness of the divide-and-conquer strategy, the comprehensiveness of experiments, and the clarity of presentation.
>
> > **Q1: Regarding the impact of image set size (n):**
>
> Thank you for the insightful question. **In summary, as the set size $n$ increases, the spatial distance between sub-image latents grows during set-aware generation, weakening cross-image interactions and causing more noticeable consistency degradation.** As discussed in Appendix F (“Consistency Degradation in Grid Layout”) and illustrated in Figure 8, compact layouts such as 2×2 or 2×3 keep latents close, allowing DiT’s cross-attention to effectively propagate identity or style cues. In contrast, elongated layouts (e.g., 1×5 or 1×7) increase latent separation, leading to issues such as drifting logos or missing fine-grained details.
>
> In current literature, most related works typically use relatively small set sizes (such as 2\~5 images in ICLora [1] and 1Propmt1Story [2]). Scaling beyond this range remains largely an open research challenge. In our work, to handle sets larger than 6 images, we adopt a sliding-window strategy with a window size of 4 and a stride of 2 （Appendix C）, enabling AutoT2IS to maintain consistency while keeping computational cost tractable. We will make this point more explicit in the main paper.
>
> For future work, we plan to explore positional-aware fusion modules and hierarchical grid attention to mitigate long-range spatial-decay and better extend AutoT2IS to larger sets (e.g., 8–10 images) without relying on windowing.
>
> [1] In-Context LoRA for Diffusion Transformers
>
> [2] One-Prompt-One-Story: Free-Lunch Consistent Text-to-Image Generation Using a Single Prompt
>
>
> > **Q2: Regarding interleaved text-and-image generation:**
>
> We appreciate this valuable suggestion. We are aware of  OpenING (Zhou et al., CVPR 2025) and have compared it with our proposed T2IS-Bench in Appendix Table 4.
>
> While both these two tasks involve multi-image reasoning, their problem settings and scopes differ substantially. Interleaved generation focuses on producing alternating text–image sequences in narrative contexts, emphasizing linguistic coherence and temporal progression. In contrast, T2IS generation produces a set of images governed purely by visual-consistency constraints（identity, style, logic), without intermediate textual steps. Moreover, the scope of T2IS-Bench spans diverse consistency-related subtasks that go beyond the narrative-centric setting of interleaved generation. We will further expand this discussion and explicitly reference InterleavedEval (EMNLP 2024) and related works in the revision to clarify their distinctions and relationships.

---

### Author Response · Authors · 2025-12-02
**Discussion Summary**

Dear Area Chair,

Thanks for your effort! To assist your assessment and reduce your workload following the recent system reset, we summarize below the key concerns and the major clarifications/experiments provided during the rebuttal period.

**Prior to the rebuttal, three out of four reviewers (g4bk, ZBFL, uWDA) gave positive ratings (6), recognizing the novelty of the Text-to-ImageSet (T2IS) task, the comprehensive experimentation, and the effectiveness of the proposed method, AutoT2IS.**

> **1. Core Research Significance and Task Novelty.**

**Concern:** The primary concern, raised by Reviewers uWDA and H1wC, centered on the significance of T2IS: how it differs from aggregating existing benchmarks and what new research challenges it unlocks.

**Action:** We clarified that existing datasets lack complete coverage of consistency dimensions. Similar to how NLP unified heterogeneous tasks to expose generalization behaviors that enabled the rise of LLMs, T2IS-Bench fills a unique gap: it is the first benchmark explicitly designed to evaluate general visual consistency.

**Reviewer uWDA explicitly confirmed this concern was adequately addressed in their follow-up response.**

> **2. Validation of T2IS-Eval Reliability.**

**Concern:** Reviewers H1wC and ZBFL raised points regarding the reliability of our MLLM-based evaluation framework, T2IS-Eval.

**Action:** We supplemented our analysis with comprehensive human evaluation studies. Results demonstrated that T2IS-Eval substantially outperforms widely used metrics such as CLIP-I and DreamSim, achieving a strong correlation with human judgments, providing a scalable and validated metric for a previously unstandardized task.

> **3. Clarification of Methodological Novelty.**

**Concern:** Concerns were raised by Reviewer H1wC about the novelty of the Structured Recaption module.

**Action:** We clarified that Structured Recaption serves a practical purpose: providing structured, disentangled, and stable prompts to enable the subsequent,  innovative Set-Aware Generation component. The core contribution lies in this divide-and-conquer strategy, which maximally leverages Diffusion Transformers' in-context capabilities.

> **4. Summary of Additional Experiments and Revisions:**

To address all other concerns, we conducted extensive analysis and revised the manuscript.

- **Quantitative Ablations (Reviewer ZBFL):** We provided extensive ablations using different image generation backbones (e.g., FLUX.1, Qwen-Image, Hunyuan-Image 3.0). The results confirmed that AutoT2IS robustly enhances set-level consistency regardless of the backbone, validating its generality.

- **Set Size Analysis (Reviewer g4bk):** We analyzed the impact of image set size $n$, and explained our sliding-window strategy for handling larger sets.

- **Related Work (Reviewer g4bk):** We expanded the discussion to better distinguish T2IS from interleaved text-and-image generation benchmarks (e.g., OpenING).

- **Training Utility (Reviewer uWDA):** We clarified how T2IS-Bench enables the training of generalized models via high-quality data synthesis (SFT) and fine-grained reward modeling (RL).

- **Manuscript Polish:** We have agreed to replace the teaser image to ensure it shows correct, representative results (Reviewer uWDA) and explained performance gaps in specialized subcategories (Reviewer ZBFL).

**We have uploaded a Revised Manuscript incorporating all these clarifications, related works, and supplementary experiments. Thank you once again for your effort and dedication.**

---

### Note · Program_Chairs · 2026-01-17
**Submission Desk Rejected by Program Chairs**

The following references in this submission do not refer to real documents and/or have major errors in bibliographic information:

 Jason Baldridge, Kevin Shih, Yael Li, Zongyi Wang, Shweta Prabhudesai, Sashi Cheluvaraju, Xin Li, Lucy Chai, Miao Ding, Bryan Catanzaro, et al. Imagen 2: Tuning text-to-image diffusion models for photorealism and generalization. arXiv preprint arXiv:2401.18680, 2024.